# Multi-stage Episodic Control for Strategic Exploration in Text Games

**Jens Tuyls[1], Shunyu Yao[1], Sham Kakade[2] & Karthik Narasimhan[1]**
[1]Department of Computer Science, Princeton University
[2]John A. Paulson School of Engineering and Applied Sciences, Harvard University
`{jtuyls, shunyuy, karthikn}`@princeton.edu, `sham@seas.harvard.edu`

## Abstract

Text adventure games present unique challenges to reinforcement learning methods due to their combinatorially large action spaces and sparse rewards. The interplay of these two factors is particularly demanding because large action spaces require extensive exploration, while sparse rewards provide limited feedback. This work proposes to tackle the explore-vs-exploit dilemma using a multi-stage approach that explicitly disentangles these two strategies within each episode. Our algorithm, called eXploit-Then-eXplore (`XTX`), begins each episode using an exploitation policy that imitates a set of promising trajectories from the past, and then switches over to an exploration policy aimed at discovering novel actions that lead to unseen state spaces. This policy decomposition allows us to combine global decisions about which parts of the game space to return to with curiosity-based local exploration in that space, motivated by how a human may approach these games. Our method significantly outperforms prior approaches by 27% and 11% average normalized score over 12 games from the Jericho benchmark (Hausknecht et al., 2020) in both deterministic and stochastic settings, respectively. On the game of Zork1, in particular, `XTX` obtains a score of 103, more than a 2x improvement over prior methods, and pushes past several known bottlenecks in the game that have plagued previous state-of-the-art methods.[1]

## 1 Introduction

Text adventure games provide a unique test-bed for algorithms that integrate reinforcement learning (RL) with natural language understanding. Aside from the linguistic ingredient, a key challenge in these games is the combination of very large action spaces with sparse rewards, which calls for a delicate balance between exploration and exploitation. For instance, the game of Zork1 can contain up to fifty *valid* action commands per state[2] to choose from. Importantly, unlike other RL environments (Bellemare et al., 2013; Todorov et al., 2012), the set of valid action choices does not remain constant across states, with unseen actions frequently appearing in later states of the game. For example, Figure 1 shows several states from Zork1 where a player has to issue unique action commands like '*kill troll with sword*', '*echo*' or '*odysseus*' to progress further in the game. This requires a game-playing agent to perform extensive exploration to determine the appropriateness of actions, which is hard to bootstrap from previous experience. On the other hand, since rewards are sparse, the agent only gets a few high-scoring trajectories to learn from, requiring vigorous exploitation in order to get back to the furthest point of the game and make progress thereon. Prior approaches to solving these games (He et al., 2016a; Hausknecht et al., 2020; Ammanabrolu & Hausknecht, 2020; Guo et al., 2020) usually employ a single policy and action selection strategy, making it difficult to strike the right balance between exploration and exploitation.

In this paper, we propose eXploit-Then-eXplore (`XTX`), an algorithm for multi-stage control to explicitly decompose the exploitation and exploration phases within each episode. In the first phase, the agent selects actions according to an *exploitation policy* which is trained using self-imitation

---

[1]Our code is available at `https://github.com/princeton-nlp/XTX`
[2]Valid actions are a feature in the Jericho simulator (Hausknecht et al., 2020) to improve computational tractability for RL. Without this handicap, the number of possible action commands is almost 200 billion.

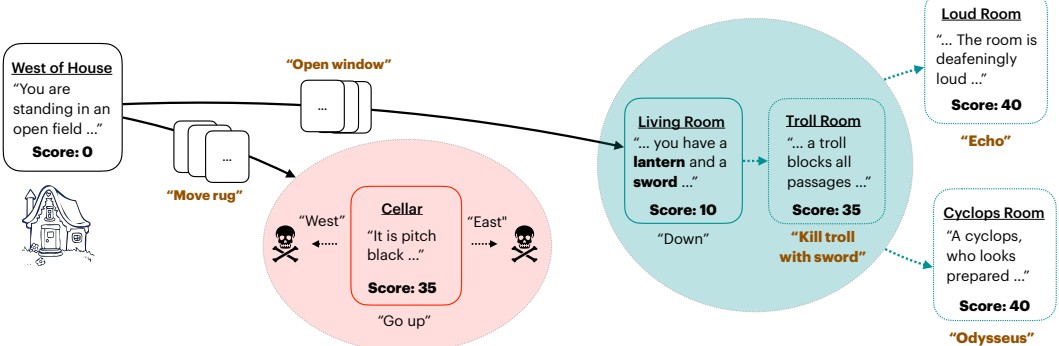

**Figure 1:** Sample game paths and state observations from ZORK1. Starting from the leftmost state ('West of House'), the agent encounters several novel and unique valid actions (e.g *Odysseus*, *Echo*) (in brown) across different states in the game. In order to make progress, our algorithm (XTX) strategically re-visits different frontiers in the state space (red and blue circles) and performs strategic local exploration to overcome bottleneck states (e.g. 'Troll Room') and dead-ends (e.g. 'Cellar'). Solid borders indicate visited states, dotted ones indicate potential future states.

learning on a mixture of promising trajectories from its past experience sampled using a combination of factors such as episodic scores and path length. This policy allows the agent to return to a state at the frontier of the state space it has explored so far. Importantly, we ensure that this policy is trained on a mixture of trajectories with different scores, in order to prevent the agent from falling into a local minimum in the state space (e.g. red space in Figure 1). In the second phase, an *exploration policy* takes over and the agent chooses actions using a value function that is trained using a combination of a temporal difference (TD) loss and an auxiliary inverse dynamics loss (Pathak et al., 2017). This allows the agent to perform strategic exploration around the frontier by reusing values of previously seen actions while exploring novel ones in order to find rewards and make progress in the game. To allow for more fine-grained control, we use a mixture of policies for both exploration and exploitation, and only change a single interpolation parameter to switch between phases.

The two-stage approach to gameplay in XTX allows an agent to combine global decisions about which parts of the game space to advance, followed by local exploration of sub-strategies in that space. This is similar to how humans tackle these games: if a player were to lose to a troll in the dungeon, they would immediately head back to the dungeon after the game restarts and explore strategies thereon to try and defeat the troll. XTX's multi-stage episodic control differs from prior approaches that add exploration biases to a single policy through curiosity bonuses (Pathak et al., 2017; Tang et al., 2017) or use different reward functions to train a separate exploration policy (Colas et al., 2018; Schäfer et al., 2021; Whitney et al., 2021). Moreover, in contrast to methods like Go-Explore (Ecoffet et al., 2021; Madotto et al., 2020), XTX does not have global phases of random exploration followed by learning —instead, both our policies are continuously updated with new experience, allowing XTX to adapt and scale as the agent goes deeper into the game. XTX also does not make any assumptions about the environment being deterministic, and does not require access to underlying game simulator or additional memory archives to keep track of game trees.

We evaluate XTX on a set of games from the Jericho benchmark (Hausknecht et al., 2020), considering both deterministic and stochastic variants of the games. XTX outperforms competitive baselines on all 12 games, and achieves an average improvement of 5.8% in terms of normalized scores across all games. For instance, on Zork1, our method obtains a score of 103 in the deterministic setting and 67 in the stochastic setting — substantial improvements over baseline scores of 44 and 41, respectively. We also perform ablation studies to demonstrate the importance of the multi-stage approach, as well as several key design choices in our exploitation and exploration policies.

## 2 RELATED WORK

**Reinforcement learning for text-based games** Prior work on building autonomous agents for text adventure games has explored several variants of reinforcement learning (RL) agents equipped with a language understanding module (see Osborne et al. (2021) for a detailed survey). Innovations on

the language representation side include using deep neural networks for handling text sequences trained using RL (Narasimhan et al., 2015; He et al., 2016a), knowledge graphs to track states across trajectories (Ammanabrolu & Hausknecht, 2020; Adhikari et al., 2020; Xu et al., 2020), and incorporating question answering or reading comprehension modules (Ammanabrolu et al., 2020; Guo et al., 2020). While these approaches focus mainly on the issues of partial observability and language semantics, they all suffer from challenges due to the large action space and sparse rewards found in games from benchmarks like Jericho (Hausknecht et al., 2020). Some approaches aim to navigate the large action space by filtering inadmissible actions (Zahavy et al., 2018; Jain et al., 2020), leveraging pre-trained language models for action selection (Yao et al., 2020; Jang et al., 2020) or word embeddings for affordance detection (Fulda et al., 2017). Recent work has also explored tackling sparse rewards by employing hierarchical policies (Xu et al., 2021).

**Navigating the exploration-exploitation trade-off in RL** The trade-off between exploration and exploitation is a well-known issue in RL (Sutton & Barto, 2018; François-Lavet et al., 2018; Kearns & Singh, 2002; Brafman & Tennenholtz, 2002). In this respect, we can broadly categorize prior techniques into two types. The first type includes methods with mixed objectives that balance exploration with exploitation. Oh et al. (2018) introduced the idea of self-imitation learning on high-scoring episodes to exploit good trajectories, as an auxiliary objective to standard actor-critic methods. Prior work has also explored the addition of curiosity bonuses to encourage exploration (Pathak et al., 2017; Tang et al., 2017; Li et al., 2020; Bellemare et al., 2016; Machado et al., 2020; Taiga et al., 2021). While we leverage self-imitation learning for exploitation and inverse dynamics bonuses for exploration, we use a multi-stage mixed policy. Other works learn a mixture of policies for decoupling exploration and exploitation, either by using a conditional architecture with shared weights (Badia et al., 2020), pre-defining an exploration mechanism for restricted policy optimization (Shani et al., 2019), or learning separate task and exploration policies to maximize different reward functions (Colas et al., 2018; Schäfer et al., 2021; Whitney et al., 2021). While we also train multiple policies, our multi-stage algorithm performs distinct exploitation and exploration phases within each episode, not requiring pre-defined exploration policies or phases. Further, we consider environments with significantly larger action spaces that evolve dynamically as the game progresses.

The second class of algorithms explicitly separate exploitation and exploration in each episode. Methods like $E^3$ (Kearns & Singh, 2002; Henaff, 2019) maintain a set of dynamics models to encourage exploration. Policy-based Go-Explore (Ecoffet et al., 2021) uses self-imitation learning to 'exploit' high-reward trajectories, but requires choosing intermediate sub-goals for the agent to condition its policy on. PC-PG (Agarwal et al., 2020) uses a policy cover to globally choose state spaces to return to, followed by random exploration. Compared to these approaches, we perform more strategic local exploration due to the use of a Q-function with inverse dynamics bonus and do not require any assumptions about determinism or linearity of the MDP. We provide a more technical discussion on the novelty of our approach at the end of Section 3.

**Directed exploration in text-based games** As previously mentioned, the large dynamic action space in text games warrant specific strategies for directed exploration. Ammanabrolu et al. (2020) used a knowledge-graph based intrinsic motivation reward to encourage exploration. Jang et al. (2020) incorporated language semantics into action selection for planning using MCTS. Both methods utilize the determinism of the game or require access to a simulator to restart the game from specific states. Madotto et al. (2020) modified the Go-Explore algorithm to test generalization in the CoinCollector (Yuan et al., 2018) and Cooking world domains (Côté et al., 2018). Their method has two phases — the agent first randomly explores and collects trajectories and then a policy is learned through imitation of the best trajectories in the experience replay buffer. In contrast, our algorithm provides for better exploration of new, unseen actions in later stages of the game through the use of an inverse dynamics module and performs multiple rounds of imitation learning for continuous scaling to deeper trajectories in the game. Recently, Yao et al. (2021) used inverse dynamics to improve exploration and Yao et al. (2020) used a language model to generate action candidates that guide exploration. However, both approaches did not employ a two-stage rollout like our work, and the latter considers a different setup without any valid action handicap.

## 3 METHOD

**Background** Text-adventure games can be formalized as a Partially Observable Markov Decision Process (POMDP) $\langle S, T, A, O, R, \gamma \rangle$. The underlying state space $S$ contains all configurations of the game state within the simulator, which is unobserved by the agent. The agent receives observations from $O$ from which it has to infer the underlying state $s \in S$. The action set $A$ consists of short phrases from the game vocabulary, $T(s'|s, a)$ is the transition function which determines the probability of moving to the next state $s'$ given the agent has taken action $a$ in state $s$, $R(s, a)$ determines the instantaneous reward, and $\gamma$ is the reward discount factor.

Existing RL approaches that tackle these games usually learn a value function using game rewards. One example is the Deep Reinforcement Relevance Network (DRRN) (He et al., 2016b) which trains a deep neural network with parameters $\phi$ to approximate $Q_\phi(o, a)$. This model encodes each observation $o$ and action candidate $a$ using two recurrent networks $f_o$ and $f_a$ and aggregates the representations to derive the Q-value through an MLP $q$: $Q_\phi(o, a) = q(f_o(o), f_a(a))$. The parameters $\phi$ of the model are trained by minimizing the temporal difference (TD) loss on tuples $(o, a, r, o')$ of observation, action, reward and the next observation sampled from an experience replay buffer:

$$\mathcal{L}_{\text{TD}}(\phi) = (r + \gamma \max_{a' \in A} Q_\phi(o', a') - Q_\phi(o, a))^2 \tag{1}$$

The agent samples actions using a softmax exploration policy $\pi(a|o; \phi) \propto \exp(Q_\phi(o, a))$.

**Challenges** There are two unique aspects of text adventure games that make them challenging. First, the action space is combinatorially large – usually games can accept action commands of 1-4 words with vocabularies of up to 2257 words. This means an agent potentially has to choose between $\mathcal{O}(2257^4) = 2.6 * 10^{13}$ actions at each state of the game. To make this problem more tractable, benchmarks like Jericho (Hausknecht et al., 2020) provide a valid action detector that filters out the set of inadmissible commands (i.e. commands that are either unrecognized by the game engine or do not change the underlying state of the game). However, this still results in the issue of *dynamic* action spaces for the agent that change with the state. For example, the action '*echo*' is unique to the Loud Room (see Figure 1). Second, these games have very sparse rewards (see Appendix A.1) and several bottleneck states (Ammanabrolu et al., 2020), making learning difficult for RL agents that use the same policy for exploration and exploitation (He et al., 2016a; Hausknecht et al., 2020; Ammanabrolu & Riedl, 2019). This results in issues of derailment (Ecoffet et al., 2021), with agents unable to return to promising parts of the state space, resulting in a substantial gap between average episode performance of the agent and the maximum score it sees in the game (Yao et al., 2020).

### 3.1 OUR ALGORITHM: eXPLOIT-THEN-eXPLORE (XTX)

We tackle the two challenges outlined above using a multi-stage control policy that allows an agent to globally pick promising states of the game to visit while allowing for strategic local exploration thereafter. To this end, we develop eXploit-Then-eXplore, where an agent performs the following two distinct phases of action selection in each episode.

**PHASE 1: Exploitation** In the exploitation phase, the agent makes a global decision about revisiting promising states of the game it has seen in its past episodes. We sample $k$ trajectories from the experience replay $\mathcal{D}$ using a combination of factors such as game score and trajectory length. These trajectories are then used to learn a policy cover $\pi_{\text{exploit}}$ using self-imitation learning (Oh et al., 2018) (see Section 3.2 for details). The agent then samples actions from $\pi_{\text{exploit}}$ until it has reached either (1) the maximum score seen during training or (2) a number of steps in the episode equal to the longest of the $k$ sampled trajectories. The second condition ensures the agent can always return to the longest of the $k$ sampled trajectories[3]. Once either of the above conditions is satisfied, the agent transitions to the exploration phase, adjusting its policy.

**PHASE 2: Exploration** In the exploration phase, the agent uses a different policy $\pi_{\text{explore}}$ trained using both a temporal difference loss and an auxiliary inverse dynamics bonus (Pathak et al., 2017; Yao et al., 2021) (see Section 3.3 for details). The intuition here is that the exploitation policy $\pi_{\text{exploit}}$ in phase 1 has brought the agent to the game frontier, which is under-explored and may

---

[3]It is possible that the agent needs less steps to return to a particular part of the game space and hence wastes some steps, but we empirically didn't find this to be a problem.

contain a combination of common (e.g. *"open door"*) and novel (e.g. *"kill troll with sword"*) actions. Therefore, using a combination of Q-values and inverse dynamics bonus enables the agent to perform strategic, local exploration to expand the frontier and discover new rewarding states. The agent continues sampling from $\pi_{\text{explore}}$ until a terminal state or episode limit is reached.

### 3.1.1 MIXTURE OF POLICIES FOR FINE-GRAINED CONTROL

While one could employ two completely disjoint policies for the exploitation and exploration phases, we choose a more general approach of having a policy mixture, with a single parameter $\lambda$ that can be varied to provide more fine-grained control during the two phases. Specifically, we define:

$$\pi_\lambda(a|c, o; \theta, \phi, \xi) = \lambda \pi_{\text{inv-dy}}(a|o; \theta, \phi) + (1 - \lambda) \pi_{\text{il}}(a|c; \xi). \qquad (2)$$

Here, $\pi_{\text{inv-dy}}$ refers to an exploration-inducing policy trained using TD loss with an inverse-dynamics bonus (Section 3.3) and is parameterized by $\theta$ and $\phi$. $\pi_{\text{il}}$ refers to an exploitation-inducing policy trained through self-imitation learning (Oh et al., 2018) (Section 3.2) and is parameterized by $\xi$. Note that the action distribution over actions $a$ induced by $\pi_{\text{inv-dy}}$ is conditioned only on the current observation $o$, while the one induced by $\pi_{\text{il}}$ is conditioned on context $c$ which is an augmentation of $o$ with past information. We can observe that $\lambda$ provides a trade-off between exploration (high $\lambda$) and exploitation (low $\lambda$). In our experiments, we choose a small, dynamic value, $\lambda = \frac{1}{2*T}$ for exploitation (where $T$ is episode limit) and $\lambda = 1$ for exploration. As we demonstrate later (Section 4.2), the non-zero $\lambda$ in exploitation is critical for the agent to avoid getting stuck in regions of local minima (e.g. Zork1). We now describe the individual components of the mixture.

### 3.2 IMITATION LEARNING FOR BUILDING A GLOBAL POLICY COVER ($\pi_{\text{il}}$)

We parameterize the imitation policy $\pi_{\text{il}}$ using a Transformer model (Vaswani et al., 2017) based on the GPT-2 architecture (Radford et al., 2019) that takes in a context $c = [a_{t-2}; a_{t-1}; o_t]$, i.e. the concatenation of two most recent past actions along with the current observation separated by [SEP] tokens, and outputs a sequence of hidden representations $h_0, \ldots, h_m$ where $m$ is the number of tokens in the sequence. $h_m$ is then projected to vocabulary size by multiplication with the output embedding matrix, after which softmax is applied to get probabilities for the next action token. Inspired by (Yao et al., 2020), the GPT-2 model is trained to predict the next action $a_t$ given the context $c$ using a language modeling objective (Equation 5). The training data consists of $k$ trajectories sampled from an experience replay memory $\mathcal{D}$ which stores transition tuples $(c_t, a_t, r_t, o_{t+1}, \text{terminal})$.

**Sampling trajectories** Let us define a trajectory $\tau$ as a sequence of observations, actions, and rewards, i.e. $\tau = o_1, a_1, r_1, o_2, a_2, r_2 \ldots, o_{l+1}$, where $l_\tau$ denotes the trajectory length (i.e. number of actions) and thus $l \leq T$ where $T$ is the episode limit. We sample a trajectory from $\mathcal{D}$ using a two-step process. First, we sample a score $u$ from a categorical distribution:

$$P(u) \propto \exp\left(\beta_1(u - \mu_{\mathcal{U}})/\sigma_{\mathcal{U}}\right), \quad u \in \mathcal{U} \qquad (3)$$

where $\mathcal{U}$ is the set of all unique scores encountered in the game so far, $\mu_{\mathcal{U}}$ is the mean of the set $\mathcal{U}$, and $\sigma_{\mathcal{U}}$ is its standard deviation. $\beta_1$ is the temperature and determines how biased the sampling process is towards high scores. The second step collects all trajectories with the sampled score $u$ and samples a trajectory $\tau$ based on the trajectory length $l_\tau$:

$$P(\tau \mid u) \propto \exp\left(-\beta_2(l_\tau - \mu_{\mathbb{L}_u})/\sigma_{\mathbb{L}_u}\right), \quad l_\tau \in \mathbb{L}_u \qquad (4)$$

where $\mathbb{L}_u$ is the multiset of trajectory lengths $l_\tau$ with score $u$, $\mu_{\mathbb{L}_u}$ is the mean of the elements in $\mathbb{L}_u$, and $\sigma_{\mathbb{L}_u}$ is its standard deviation. $\beta_2$ defines the temperature and determines the strength of the bias towards shorter length trajectories. We perform this sampling procedure $k$ times (with replacement) to obtain a trajectory buffer $\mathcal{B}$ on which we perform imitation learning. This allows the agent to globally explore the game space by emulating promising experiences from its past, with a bias towards trajectories with high game score and shorter lengths. The motivation for sampling shorter length trajectories among the ones that reach the same score is because those tend to be the ones that waste less time performing meaningless actions (e.g. "pick up sword", "drop sword", etc.).

**Learning from trajectories** Given the trajectory buffer $\mathcal{B}$ containing single-step ($c$, $a$) pairs of context $c$ and actions $a$ from the trajectories sampled in the previous step, we train $\pi_{\text{il}}$ by minimizing the cross-entropy loss over action commands (Yao et al., 2020):

$$\mathcal{L}(\xi) = -\mathbb{E}_{(c,a)\sim\mathcal{B}} \log \pi_{\text{il}}(a|c; \xi), \qquad (5)$$

where $c$ defines the context of past observations and actions as before, and $\xi$ defines the parameters of the GPT-2 model. We perform several passes of optimization through the trajectory buffer $\mathcal{B}$ until convergence[4] and periodically perform this optimization every $n$ epochs in gameplay to update $\pi_{\text{il}}$. Furthermore, $\pi_{\text{il}}$ is renormalized over the valid action set $A_v$ during gameplay. Note that while $\pi_{\text{il}}$ is trained similarly to the GPT-2 model in (Yao et al., 2020), their model *generates* action candidates.

### 3.3 EFFICIENT LOCAL EXPLORATION WITH INVERSE DYNAMICS ($\pi_{\text{inv}-\text{dy}}$)

In the second phase of our algorithm, we would like to use a policy that tackles (1) the large action space and (2) the dynamic nature of the action set at every step in the game, which makes it crucial to keep trying under-explored actions and is difficult for the Q network alone to generalize over. To this end, we use the inverse dynamics model (INV-DY) from (Yao et al., 2021). INV-DY is a Q-based policy $\pi_{\text{inv}-\text{dy}}$ similar to DRRN (He et al., 2016a), optimized with the standard TD loss (see Background of Section 3). In addition, it adds an auxiliary loss $\mathcal{L}_{\text{inv}}$ capturing an inverse dynamics prediction error (Pathak et al., 2017), which is added as an intrinsic reward to the game reward ($r = r_{\text{game}} + \alpha_1 * \mathcal{L}_{\text{inv}}$) and hence incorporated into the TD loss. Formally, $\mathcal{L}_{\text{inv}}$ is defined as:

$$\mathcal{L}_{\text{inv}}(\theta, \phi) = -\log p_d(a|g_{\text{inv}}(\text{concat}(f_o(o), f_o(o')))), \qquad (6)$$

where $\theta$ denotes the parameters for the recurrent decoder $d$ and the MLP $g_{inv}$ (neither of which are used in $\pi_{\text{inv}-\text{dy}}$ during gameplay to score the actions), and $f_o$ is the encoder defined in Section 3. This loss is optimized together with the TD loss as well as with an action decoding loss $\mathcal{L}_{\text{dec}}$ to obtain the following overall objective that is used to train $\pi_{\text{inv}-\text{dy}}$:

$$\mathcal{L}(\phi, \theta) = \mathcal{L}_{\text{TD}} + \alpha_2 \mathcal{L}_{\text{inv}}(\phi, \theta) + \alpha_3 \mathcal{L}_{\text{dec}}(\phi, \theta), \qquad (7)$$

where $\mathcal{L}_{\text{dec}}(\phi, \theta) = -\log p_d(a|f_a(a))$. Here, $f_a$ is a recurrent network (see Section 3). Please refer to Yao et al. (2021) for details. We train the model by sampling batches of transitions from a prioritized experience replay buffer $\mathcal{D}$ (Schaul et al., 2015) and performing stochastic gradient descent. Inverse dynamics ameliorates the challenges (1) and (2) mentioned above by rewarding underexplored actions (i.e. a high loss in Equation 6) and by generalizing over novel action commands. Specifically, the INV-DY network might generalize through learning of past bonuses to what new actions would look like and hence *identify* novel actions before having tried them once.

### 3.4 EPISODIC ROLLOUTS WITH XTX (ALGORITHM 1)

We now describe how XTX operates in a single episode. The agent starts in phase 1, where actions $a_t$ are sampled from $\pi_{\text{exploit}}$. Following this exploitation policy brings the agent to the game frontier, which we estimate to happen when either (1) the current episode score $\geq M$, the maximum score in the trajectory buffer $\mathcal{B}$ or when (2) the current time step $t > l_{\max}$, the length of the longest trajectory in $\mathcal{B}$. The agent then enters phase 2 and switches its strategy to sample actions only from $\pi_{\text{inv}-\text{dy}}$ by setting $\lambda = 1$. At every time step $t$ during all phases, a transition tuple $(c_t, a_t, r_t, o_{t+1}, \text{terminal})$ is added to the replay buffer $\mathcal{D}$. The policy $\pi_{\text{exploit}}$ is updated every $n$ episodes using the process in Section 3.2, while $\pi_{\text{explore}}$ is updated within episodes at every step using the TD loss in equation 7, sampling high rewarding trajectories with priority fraction $\rho$, similar to (Guo et al., 2020)[5].

### 3.5 NOVELTY IN COMPARISON TO PRIOR ALGORITHMS

We now more explicitly discuss comparisons to a few other approaches. The most closely related approaches are multi-stage algorithms, including Go-Explore (Ecoffet et al., 2021) and PC-PG (Agarwal et al., 2020) (and somewhat the $E^3$ algorithm (Kearns & Singh, 2002; Henaff, 2019)). Both of these algorithms can be viewed as approaches which explicitly use a "roll-in" policy, with the goal of visiting a novel region of the state-action space. Go-Explore is limited to deterministic MDPs (where it is easy to re-visit any state in the past), while PC-PG (applicable to more general MDPs with provable guarantees under certain linearity assumptions) more explicitly builds a set of policies ('policy cover') capable of visiting different regions of the state space. However, in both of these

---

[4]We empirically find 40 passes to be sufficient for convergence.

[5]We slightly differ from their approach as we only prioritize transitions from trajectories that achieve the maximum score seen so far.

---

**Algorithm 1** The eXploit-Then-eXplore (XTX) algorithm

---

1: Initialize prioritized replay memory $\mathcal{D}$ to capacity N with priority fraction $\rho$.
2: Initialize $\pi_{\text{exploit}}$ (with parameters $\xi$) and $\pi_{\text{explore}}$ (with parameters $\theta$ and $\phi$).
3: Set max score $M$, max length $l_{\max}$ in $\mathcal{B}$ to 0.
4: Exploration steps $R = 50$; episode limit $T = 50$
5: **for** $episode \leftarrow 1, \ldots, E$ **do**
6:     **for** $t \leftarrow 1, \ldots, T$ **do**
7:         Receive observation $o_t$ and valid action set $A_v \subset A$ for current state.
8:         **if** current episode score $< M$ and $t < T - R$ **then**       ▷ PHASE 1
9:             $\lambda \leftarrow \frac{1}{2*T}$
10:        **else**       ▷ PHASE 2
11:             $\lambda \leftarrow 1$
12:        **end if**
13:         Sample an action $a_t$ from policy $\pi_\lambda(a_t|o_t, a_{t-1}, a_{t-2}; \phi, \theta, \xi)$.       ▷ Equation 2
14:         Step with $a_t$ and receive $(r_t, o_{t+1}, \text{terminal})$ from game engine.
15:         Store transition tuple $(c_t, a_t, r_t, o_{t+1}, \text{terminal})$ in $\mathcal{D}$.
16:         Update $\pi_{\text{inv}-\text{dy}}$ using TD loss with inverse dynamics.       ▷ Equation 7
17:     **end for**
18:     **if** $n$ episodes have passed **then**
19:         Sample $k$ trajectories from $\mathcal{D}$ to form the new trajectory buffer $\mathcal{B}$.       ▷ Section 3.2
20:         Update $\pi_{\text{il}}$ with cross-entropy loss.       ▷ Equation 5
21:         Update $M, l_{\max}$ and set $T \leftarrow l_{\max} + R$.
22:     **end if**
23: **end for**

---

approaches, once the agent reaches a novel part of the state-space, the agent acts randomly. A key distinction in our approach is that once the agent reaches a novel part of the state space, it uses an exploration with novelty bonuses, which may more effectively select promising actions over a random behavioral policy in large action spaces.

The other broad class of approaches that handle exploration use novelty bonuses, with either a policy gradient approach or in conjunction with $Q$-learning (see Section 2). The difficulty with the former class of algorithms is the "catastrophic forgetting" effect (see Agarwal et al. (2020) for discussion). The difficulty with $Q$-learning approaches (with a novelty bonus) is that bootstrapping approaches, with function approximation, can be unstable in long planning horizon problems (sometimes referred to as the "deadly triad" (Jiang et al., 2021)). While XTX also uses $Q$-learning (with a novelty bonus), we only use this policy ($\pi_{\text{inv}-\text{dy}}$) in the second phase of the algorithm in contrast to (Yao et al., 2021) where $\pi_{\text{inv}-\text{dy}}$ is used throughout the entire episode; our hope is that this instability can be alleviated since we are effectively using $Q$-learning to solve a shorter horizon exploration problem (as opposed to globally using $Q$-learning, with a novelty bonus).

## 4 EXPERIMENTS

**Environments** We evaluate on 12 human-created games across several genres from the Jericho benchmark (Hausknecht et al., 2020). They provide a variety of challenges such as darkness, non-standard actions, inventory management, and dialog (Hausknecht et al., 2020). At every step $t$, the observation from the Jericho game engine contains a description of the state, which is augmented with location and inventory information (by issuing "look" and "inventory" commands) to form $o_t$ (Hausknecht et al., 2019). In addition, we use of the valid action handicap provided by Jericho, which filters actions to remove those that do not change the underlying game state. We found this action handicap to be imperfect for some games (marked with * in Table 1), and manually added some actions required for agent progress from game walkthroughs to the game engine's grammar.

**Evaluation** We evaluate agents under two settings: (a) a deterministic setting where the transition dynamics $T(s'|s, a)$ is a one-hot vector over all the next states $s'$ and (b) a stochastic setting[6] where the $T(s'|s, a)$ defines a distribution over next states $s'$, and the observations $o$ can be perturbed with

---

[6]Only six games have stochastic variants, and DRAGON was left out due to memory issues in the baselines.

| Games | DRRN | | INV-DY | | RC-DQN | | XTX-Uniform | | XTX (ours) | | Δ (%) | Game Max |
|---|---|---|---|---|---|---|---|---|---|---|---|---|
| | Avg | Max | Avg | Max | Avg | Max | Avg | Max | Avg | Max | | |
| ZORK1 | 40.3 | 55.0 | 44.1 | 105.0 | 41.7 | 53.0 | 34.1 | 52.3 | **103.4** | **152.7** | +17% | 350 |
| INHUMANE* | 34.8 | 56.7 | 27.7 | 63.3 | 29.8 | 53.3 | 59.2 | **76.7** | **64.0** | **76.7** | +5% | 90 |
| LUDICORP* | 17.1 | 48.7 | 19.6 | 49.3 | 10.9 | 40.7 | 67.3 | 86.0 | **78.8** | **91.0** | +8% | 150 |
| ZORK3* | 0.3 | 4.3 | 0.5 | **5.0** | 3.0 | **5.0** | 3.8 | 4.7 | **4.2** | **5.0** | +6% | 7 |
| PENTARI* | 45.6 | 58.3 | 34.5 | 53.3 | 33.4 | 46.7 | 43.4 | **60.0** | **49.6** | **60.0** | +6% | 70 |
| DETECTIVE | 289.9 | 320.0 | 289.5 | 323.3 | 269.3 | **346.7** | 296.0 | 336.7 | **312.2** | 340.0 | +4% | 360 |
| BALANCES* | 14.1 | 25.0 | 12.5 | 25.0 | 10.0 | 18.3 | 21.9 | 25.0 | **24.0** | **26.7** | +4% | 51 |
| LIBRARY* | 24.8 | **30.0** | 24.7 | **30.0** | 24.2 | **30.0** | 26.1 | **30.0** | **28.5** | **30.0** | +8% | 30 |
| DEEPHOME* | 58.8 | 68.0 | 58.9 | 72.7 | 1.0 | 1.0 | 52.6 | 70.0 | **77.7** | **92.3** | +6% | 300 |
| ENCHANTER* | 42.0 | **66.7** | 44.2 | 63.3 | 26.8 | 38.3 | 24.3 | 28.3 | **52.0** | **66.7** | +2% | 400 |
| DRAGON[7] | -3.7 | 8.0 | -2.3 | 8.7 | 3.2 | 8.0 | 40.7 | 126.0 | **96.7** | **127.0** | 0% | 25 |
| OMNIQUEST | 8.2 | 10.0 | 9.9 | **13.3** | 9.3 | 10.0 | 8.6 | 10.0 | **11.6** | **13.3** | +3% | 50 |
| **Avg. Norm Score** | 29.5% | 48.8% | 28.4% | 51.8% | 29.7% | 44.5% | 49.2% | 58.6% | 56.3% | 64.0% | 5.8% | 100% |

**Table 1:** Results on deterministic games for the best XTX model, where the inverse dynamics scaling coefficient $\alpha_1$ was tuned per game. We outperform the baselines on all 12 games, achieving an average normalized game score of 56%. * indicates actions were added to the game grammar. $\Delta$ indicates the absolute performance difference between XTX and the best baseline on **Avg** scores. Scores are averaged across 3 seeds. Baselines were rerun with the latest Jericho version.

irrelevant sentences such as *"you hear in the distance the chirping of a song bird"*. We report both the episode score average (**Avg**) over the last 100 episodes at the end of training, as well as the maximum score (**Max**) seen in any episode during training.

**Baselines** We consider four baselines. 1) **DRRN (He et al., 2016a)**: This model uses a Q-based softmax policy, i.e. $\pi \propto \exp(Q(o, a; \phi))$, parameterized using GRU encoders and decoders, and trained using the TD loss (Equation 1). 2) **INV-DY (Yao et al., 2021)**: Refer to Section 3.3. 3) **RC-DQN (Guo et al., 2020)**: This is a state-of-the-art model that uses an object-centric reading comprehension (RC) module to encode observations and output actions. The training loss is that of DRRN above, and during gameplay, the agent uses an $\epsilon$-greedy strategy. 4) **XTX-Uniform ($\sim$Go-Explore)**: Here, we replace $\pi_{\text{inv}-\text{dy}}$ with a policy that samples actions uniformly during Phase 2, keeping all other factors of our algorithm the same. This is closest to a version of the Go-Explore algorithm (Ecoffet et al., 2021) that returns to promising states and performs random exploration. However, while conceptually similar to Go-Explore, XTX-Uniform does not make use of any additional memory archives, and avoids training a goal-based policy. See Appendix A.2 for implementation details and hyperparameters.

### 4.1 RESULTS

**Deterministic games** We report results in Table 1 (refer to Appendix A.5 and A.6 for more details). Overall, XTX outperforms DRRN, INV-DY, RC-DQN, and XTX-Uniform by 27%, 28%, 27%, and 7% respectively, in terms of average normalized game score (i.e. average episode score divided by max score, averaged over all the games). Our algorithm performs particularly well on Zork1, achieving a 17% absolute improvement in average episode score and a 14% improvement in average maximum score compared to the best baseline. In particular, XTX manages to advance past several documented bottlenecks like the dark Cellar (see Figure 1) which have proved to be very challenging for existing methods (Ammanabrolu et al., 2020). While performance with XTX-Uniform is sometimes close, exploration with inverse dynamics instead of random exploration pushes past several bottlenecks present in Zork1 and leads to significant gains on Deephome, Enchanter, Omniquest, and Ludicorp, showing the potential usefulness of strategic exploration at the game frontier.

**Stochastic games** To show the robustness of XTX to stochasticity, we evaluate our agent in the stochastic setting (Table 2). XTX outperforms the baselines on 4 out of 5 games, and pushes past the maximum scores of XTX-Uniform on the same fraction. Especially impressive is the performance on Zork1, which is still higher than the state-of-the-art score in the *deterministic* setting.

### 4.2 ABLATION STUDIES

In order to evaluate the importance of the various components in XTX, we perform several ablations on a subset of the games as described below and shown in Figure 2 (more in Appendix A.3 and A.4).

---

[7]Interestingly, XTX manages to achieve a very high score on Dragon by exploiting an integer underflow bug.

| Games | DRRN | | INV-DY | | RC-DQN | | XTX-Uniform | | XTX (ours) | | Δ (%) | Game Max |
|---|---|---|---|---|---|---|---|---|---|---|---|---|
| | Avg | Max | Avg | Max | Avg | Max | Avg | Max | Avg | Max | | |
| ZORK1 | 41.3 | 55.7 | 36.9 | 85.7 | 40.3 | 53.0 | 31.2 | 48.0 | **67.7** | **143.0** | +8% | 350 |
| ZORK3* | 0.2 | 4.0 | 0.4 | 4.7 | **2.7** | 4.7 | 2.3 | 4.3 | 2.6 | **5.0** | -1% | 7 |
| PENTARI* | 38.2 | **60.0** | 37.5 | 55.0 | 33.3 | 41.7 | 38.8 | **60.0** | **47.3** | **60.0** | +12% | 70 |
| DEEPHOME* | 43.0 | 65.7 | 58.4 | 73.0 | 1.0 | 1.0 | 50.7 | 69.3 | **70.9** | **96.0** | +4% | 300 |
| ENCHANTER* | 42.0 | 56.7 | 34.5 | 53.3 | 27.1 | 43.3 | 30.2 | 45.0 | **44.8** | **58.3** | +1% | 400 |
| **Avg. Norm Score** | 18.9% | 39.0% | 19.7% | 41.5% | 20.9% | 30.5% | 24.3% | 39.1% | 31.8% | 48.9% | 4.8% | 100% |

**Table 2:** Results on stochastic games. We outperform baselines on 4 out of 5 games, with an average normalized game score of 32%. Scores are averaged across 3 seeds. Baselines were rerun with the latest Jericho version.

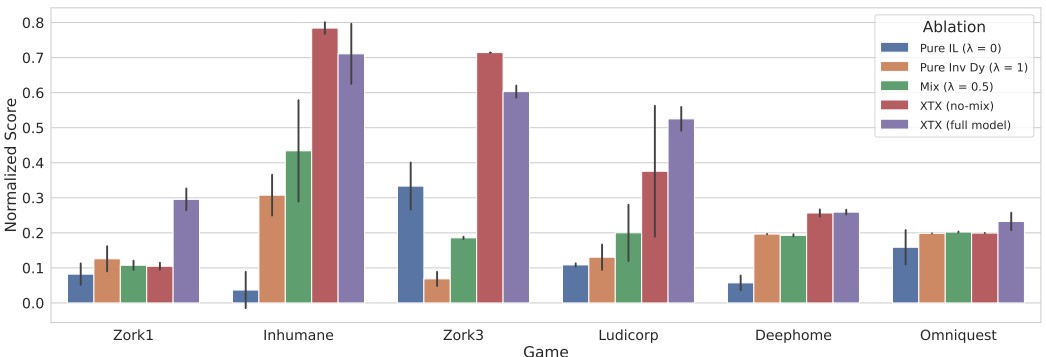

**Figure 2:** Average episode scores for 4 ablation models across 6 games. Overall, we find both the strategic inverse dynamics policy and the explicit exploitation policy to be key for our algorithm.

**Pure imitation learning** ($\lambda = 0$) This ablation sets $\lambda = 0$ in equation 2, meaning the agent will always use the imitation learning policy $\pi_{\text{il}}$. As expected, this model performs quite badly since it is based on pure exploitation and is hence unlikely to reach deep states in the game.

**Pure inverse dynamics** ($\lambda = 1$) This ablation sets $\lambda = 1$ in equation 2, hence always using the inverse dynamics exploration policy $\pi_{\text{inv}-\text{dy}}$, resulting in the model proposed in (Yao et al., 2021). While this model can sometimes achieve high maximum scores, it is unable to learn from these and hence its average episode score remains quite low, consistent with findings in (Yao et al., 2021).

**Mixing exploration and exploitation** ($\lambda = 0.5$) By setting $\lambda = 0.5$, this ablation constantly alternates between exploitation and exploration, never committing to one or the other. This causes the agent to suffer from issues of both the $\lambda = 0$ and $\lambda = 1$ models, resulting in weak results.

**Pure separation of exploitation and exploration (XTX no-mix)** In this ablation, we examine the importance of having a mixture policy in Phase 1 of the algorithm instead of setting $\lambda = 0$ in Phase 1 and to 1 in Phase 2. This explicitly separated model, denoted as XTX (no-mix), performs a bit better in the games of Inhumane and Zork3, but sometimes fails to push past certain stages in Ludicorp and completely gets stuck in the game of Zork1. This shows it is crucial to have a mixture policy in Phase 1 in order to get past bottleneck states in difficult games.

## 5 CONCLUSION

We have proposed XTX, an algorithm for multi-stage episodic control in text adventure games. XTX explicitly disentangles exploitation and exploration into different policies, which are used by the agent for action selection in different phases of the same episode. Decomposing the policy allows the agent to combine global decisions on which state spaces in the environment to (re-)explore, followed by strategic local exploration that can handle novel, unseen actions – aspects that help tackle the challenges of sparse rewards and dynamic action spaces in these games. Our method significantly outperforms prior methods on the Jericho benchmark (Hausknecht et al., 2020) under both deterministic and stochastic settings, and even surpasses several challenging bottlenecks in games like Zork1 (Ammanabrolu et al., 2020). Future work can integrate our algorithm with approaches that better leverage linguistic signals to achieve further progress in these games.

## ACKNOWLEDGEMENTS

We thank the members of the Princeton NLP group and the anonymous reviewers for their valuable feedback. JT was supported by a graduate fellowship at Princeton University. We are grateful to the Google Cloud Research program for computational support in running our experiments. We would also like to thank Matthew Hausknecht for all the help regarding the Jericho environment.

## ETHICAL CONSIDERATIONS

This work focuses on building better agents for text-adventure games and hence does not have immediate direct ethical concerns. However, the techniques introduced in this paper may be generally useful for other autonomous agents that combine sequential decision making with language understanding (e.g. dialog systems). As such agents become more capable and influential in our lives, it is important to make sure their objectives align with those of humans, and that they are free of bias.

## REPRODUCIBILITY

Our code is publicly available here `https://github.com/princeton-nlp/XTX`. We provide all implementation details such as hyperparameters, model architectures and training regimes in Appendix A.2. We used Weights & Biases for experiment tracking and visualizations to develop insights for this paper.

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

# A APPENDIX

## A.1 GAME STATISTICS

| Game | ZORK1 | INHUMANE | LUDICORP | LIBRARY | ZORK3 | PENTARI |
|---|---|---|---|---|---|---|
| **Avg./Max** | 9 / 51 | 14 / 28 | 4 / 45 | 5 / 6 | 39 / 41 | 5 / 16 |
| **Game** | DETECTIVE | BALANCES | DEEPHOME | ENCHANTER | DRAGON | OMNIQUEST |
| **Avg./Max** | 2 / 5 | 12 / 54 | 6 / 53 | 15 / 40 | 9 / 24 | 13 / 26 |

**Table 3:** Average and maximum number of steps between rewards for games in Jericho (based on human walkthroughs). Several games have long sequences of actions without reward.

Table 3 contains the average and maximum number of steps between rewards in these games, showcasing their challenging nature.

## A.2 IMPLEMENTATION DETAILS

We use a learning rate of $10^{-3}$ and $10^{-4}$ for $\pi_{\text{il}}$ and $\pi_{\text{inv}-\text{dy}}$, respectively. Both policies are trained on batches of size 64, with hidden dimensions of size 128. The scaling coefficient $\alpha_1$ for the inverse dynamics intrinsic reward is set to 1 for all games except for Deephome ($\alpha_1 = 0.1$), Enchanter ($\alpha_1 = 0.5$), Omniquest ($\alpha_1 = 2$), Ludicorp ($\alpha_1 = 0.5$), Detective ($\alpha_1 = 2$), and Pentari ($\alpha_1 = 2$). The Transformer $\pi_{\text{il}}$ has 3 layers and 4 attention heads. $\beta_1$ in equation 3 is set to 1, $\beta_2$ in equation 4 is set to 10k to encourage picking the shortest length trajectory, and $k$ is set to 10. In equation 6, $\alpha_1 = \alpha_2 = 1$. The priority fraction $\rho$ is set to 0.5. Every time $\pi_{\text{il}}$ is trained, we also scale the episode length $T$ to have at least $R$ remaining steps of exploration by setting $T = l_{\max} + R$, where $l_{\max}$ is the length of the longest trajectory in the trajectory buffer $\mathcal{B}$. In practice, $R = 50$, and hence the agent will be guaranteed at least 50 steps of exploration each episode. XTX and DRRN are run for 800k interaction steps, while RC-DQN which is run for 100k interaction steps following Guo et al. (2020).

## A.3 FULL SET OF ABLATIONS

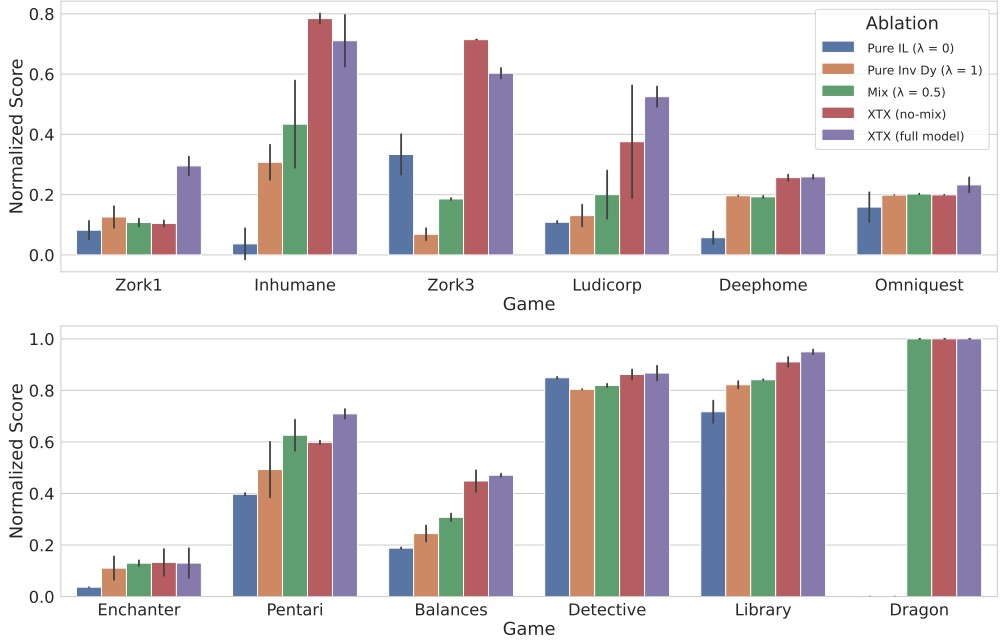

**Figure 3:** Average episode scores for 4 ablation models across 12 games. Overall, we find both the strategic inverse dynamics policy and the explicit exploitation policy to be key for our algorithm. Scores for dragon were clipped to be between 0 and 1.

A.4 ABLATION TRAINING PLOTS

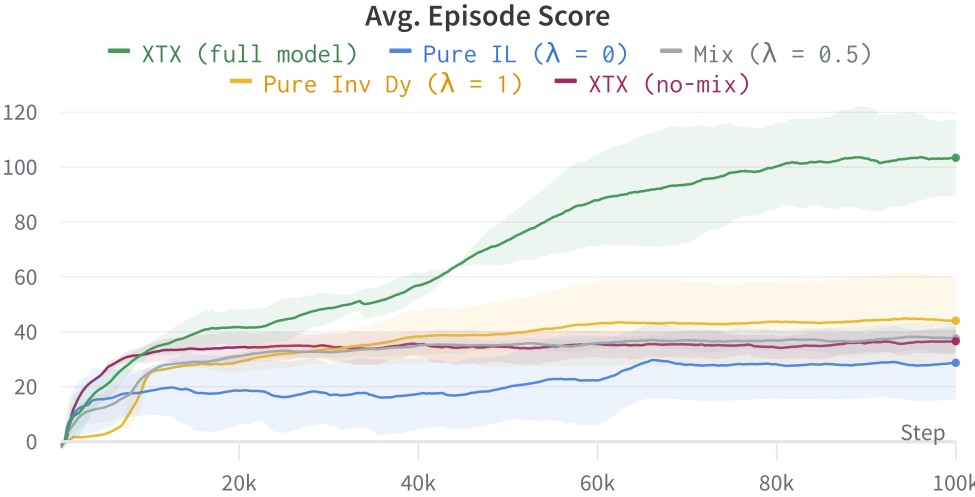

**Figure 4:** Average episode score throughout training for all ablations on Zork1. Shaded areas indicate one standard deviation.

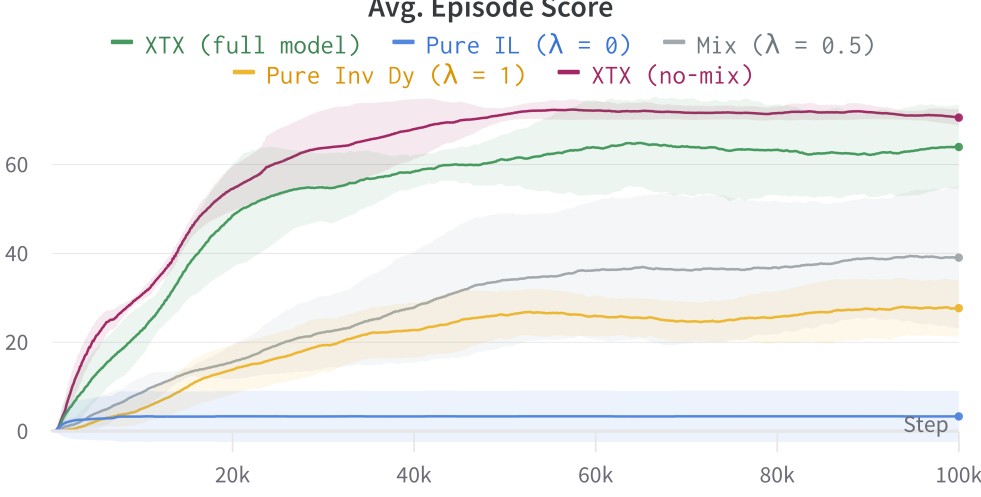

**Figure 5:** Average episode score throughout training for all ablations on Inhumane. Shaded areas indicate one standard deviation.

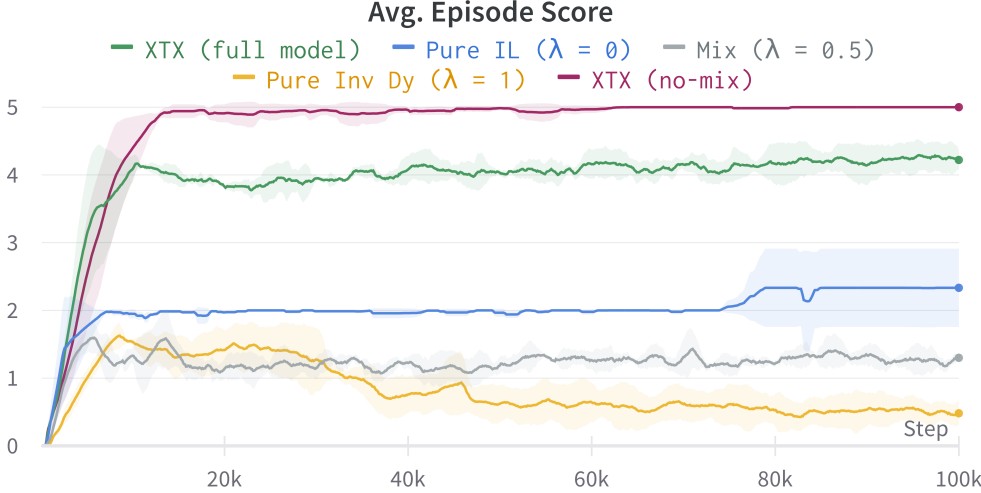

**Figure 6:** Average episode score throughout training for all ablations on Zork3. Shaded areas indicate one standard deviation.

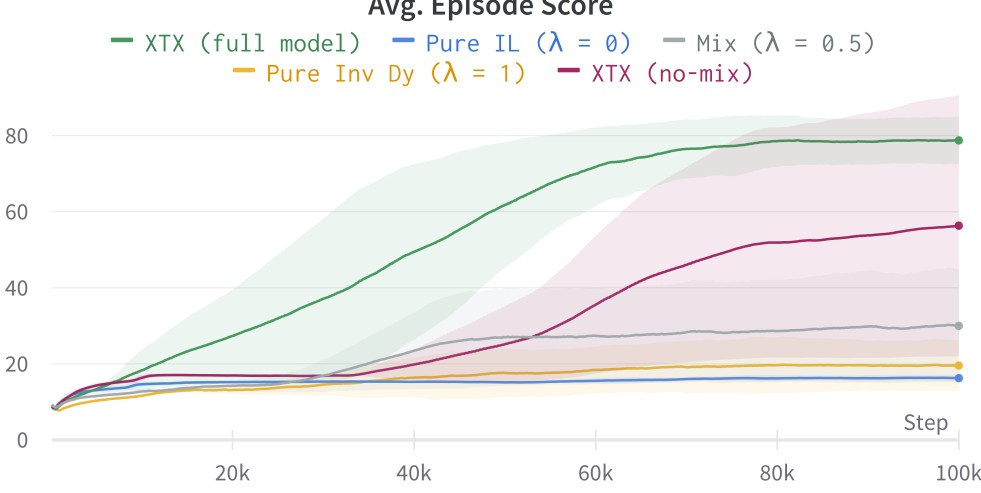

**Figure 7:** Average episode score throughout training for all ablations on Ludicorp. Shaded areas indicate one standard deviation.

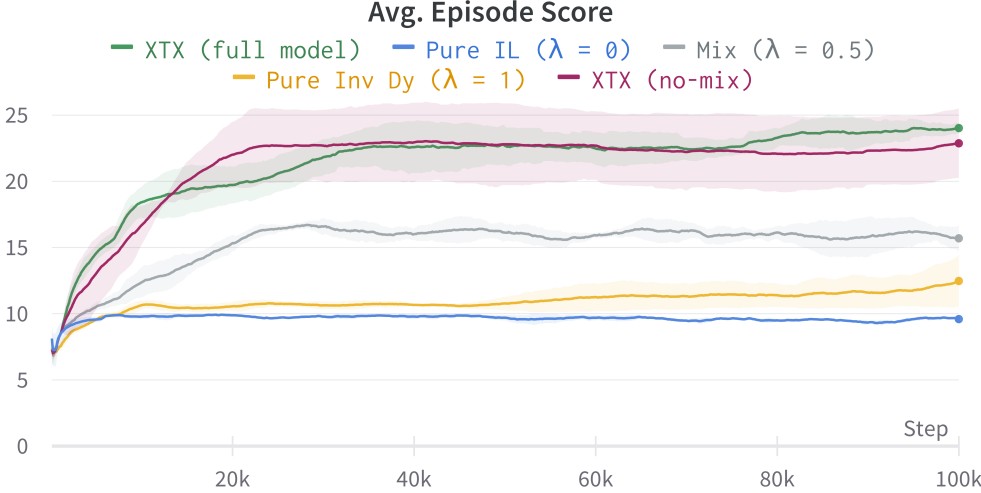

**Figure 8:** Average episode score throughout training for all ablations on Balances. Shaded areas indicate one standard deviation.

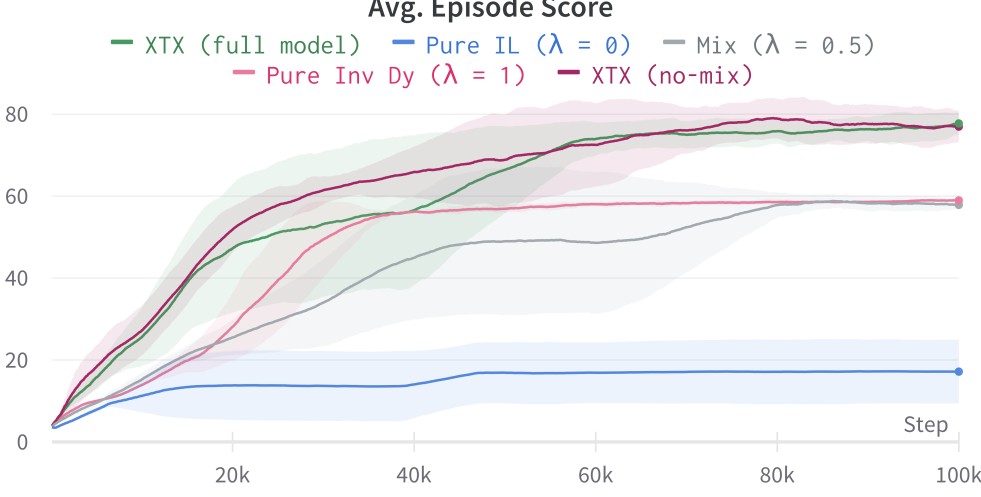

**Figure 9:** Average episode score throughout training for all ablations on Deephome. Shaded areas indicate one standard deviation.

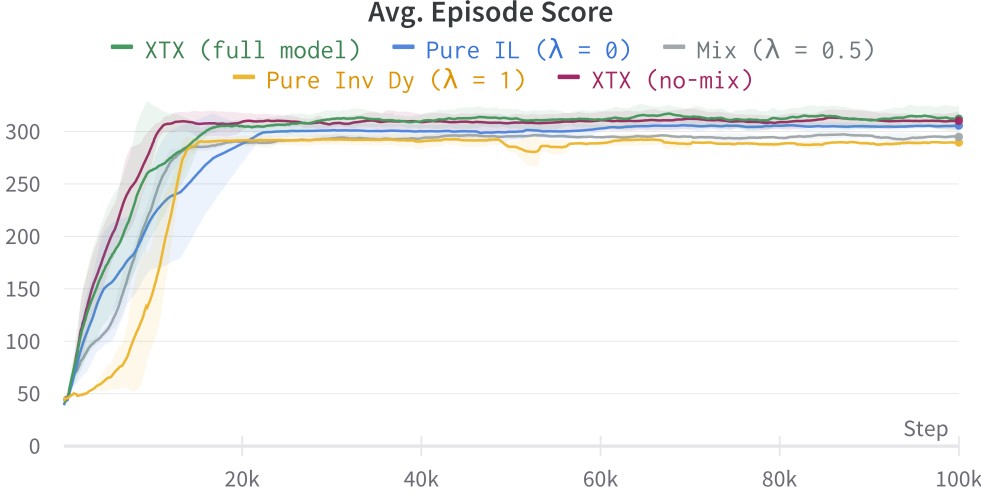

**Figure 10:** Average episode score throughout training for all ablations on Detective. Shaded areas indicate one standard deviation.

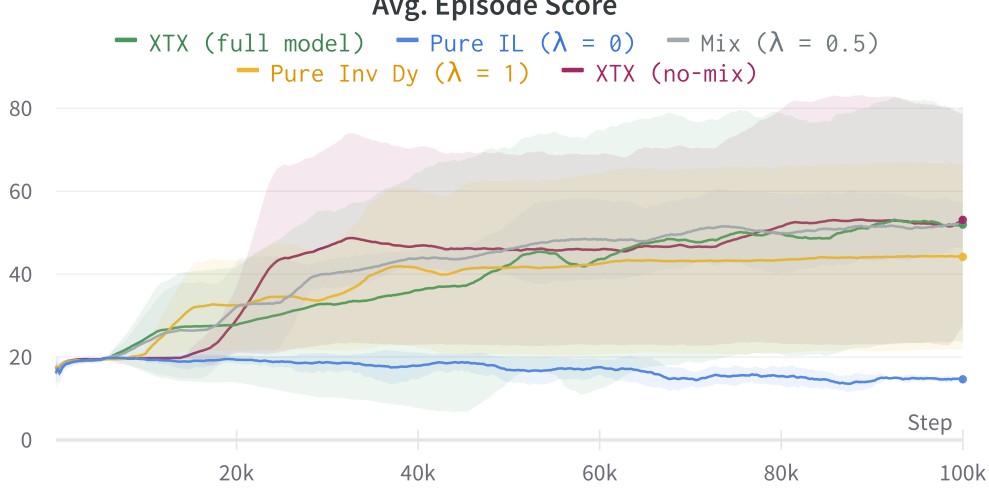

**Figure 11:** Average episode score throughout training for all ablations on Enchanter. Shaded areas indicate one standard deviation.

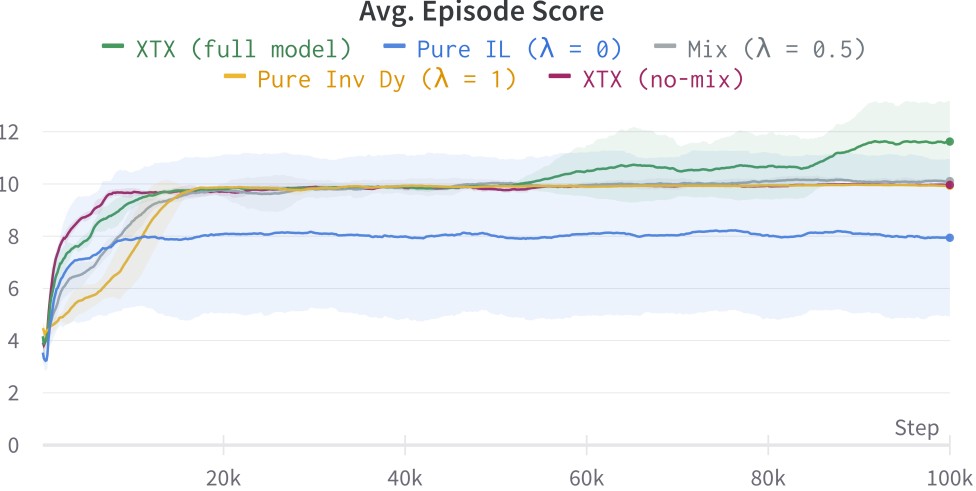

**Figure 12:** Average episode score throughout training for all ablations on Omniquest. Shaded areas indicate one standard deviation.

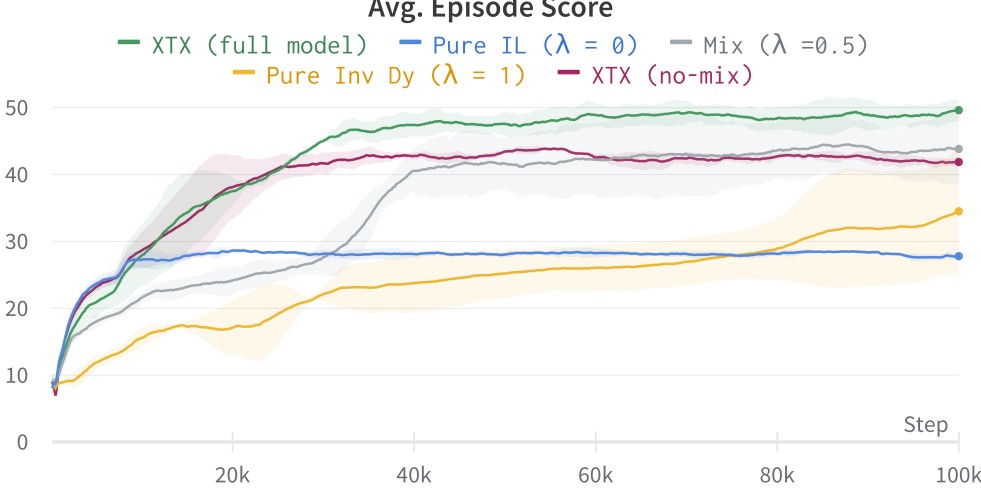

**Figure 13:** Average episode score throughout training for all ablations on Pentari. Shaded areas indicate one standard deviation.

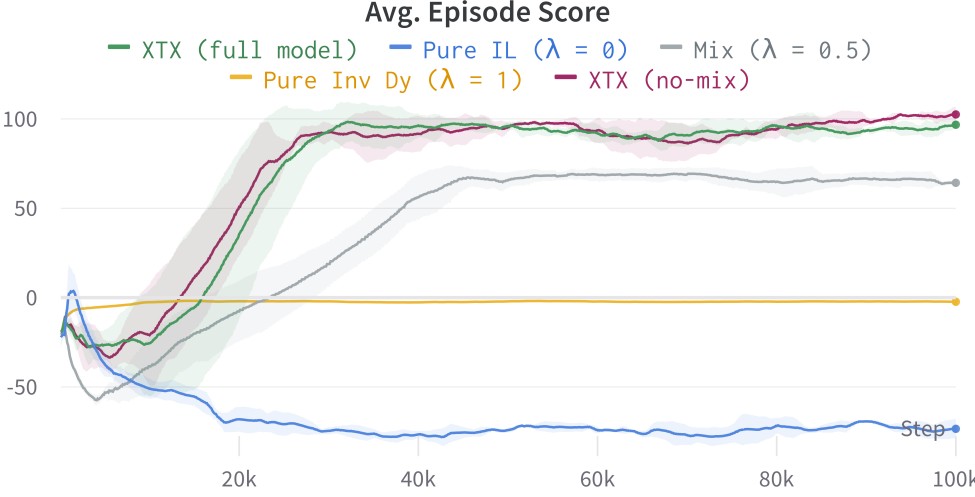

**Figure 14:** Average episode score throughout training for all ablations on Dragon. Shaded areas indicate one standard deviation.

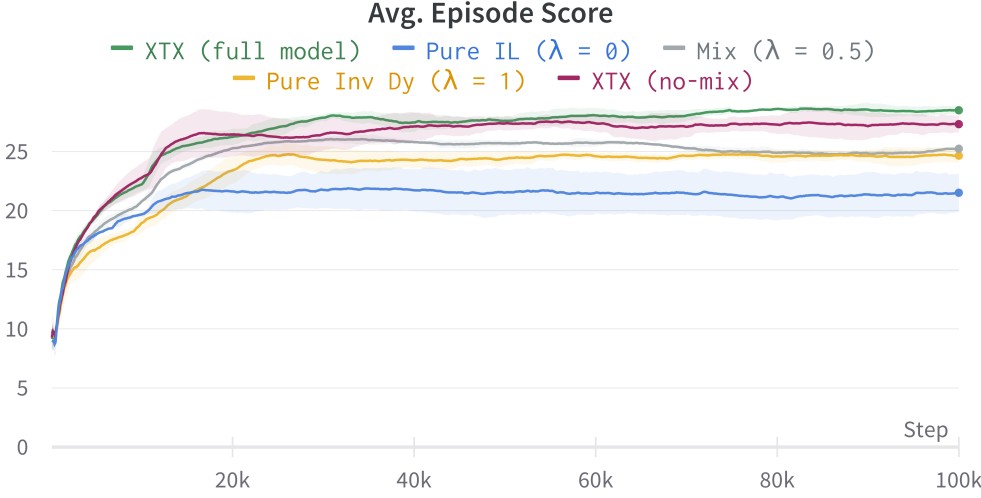

**Figure 15:** Average episode score throughout training for all ablations on Library. Shaded areas indicate one standard deviation.

## A.5 Full Deterministic and Stochastic Results

| Games | DRRN Avg | DRRN Max | INV-DY Avg | INV-DY Max | RC-DQN Avg | RC-DQN Max | XTX-Uniform Avg | XTX-Uniform Max | XTX (ours) Avg | XTX (ours) Max | Δ (%) | Game Max |
|---|---|---|---|---|---|---|---|---|---|---|---|---|
| ZORK1 | 40.3 (2.2) | 55.0 (6.4) | 44.1 (12.6) | 105.0 (19.9) | 41.7 (0.6) | 53.0 (0.0) | 34.1 (1.5) | 52.3 (3.8) | **103.4** (10.9) | **152.7** (1.7) | +17% | 350 |
| INHUMANE | 31.0 (1.0) | 56.7 (4.7) | 28.1 (3.6) | 60.0 (0.0) | 31.8 (1.4) | 63.3 (4.7) | **68.9** (8.9) | **83.3** (9.4) | 60.9 (5.9) | 70.0 (14.1) | -9% | 90 |
| INHUMANE* | 34.8 (3.9) | 56.7 (4.7) | 27.7 (5.3) | 63.3 (4.7) | 29.8 (2.3) | 53.3 (4.7) | 59.2 (1.2) | **76.7** (9.4) | **64.0** (7.7) | **76.7** (9.4) | +5% | 90 |
| LUDICORP | 15.6 (0.1) | **23.0** (0.0) | 15.6 (0.2) | **23.0** (0.0) | 12.4 (1.1) | 21.0 (2.2) | 19.9 (0.4) | **23.0** (0.0) | **20.9** (0.1) | **23.0** (0.0) | +1% | 150 |
| LUDICORP* | 17.1 (1.7) | 48.7 (2.1) | 19.6 (5.5) | 49.3 (16.2) | 10.9 (1.7) | 40.7 (2.5) | 67.3 (4.2) | 86.0 (2.8) | **78.8** (5.1) | **91.0** (3.6) | +8% | 150 |
| ZORK3 | 0.3 (0.0) | 4.7 (0.5) | 0.4 (0.0) | **5.0** (0.0) | 3.2 (0.5) | **5.0** (0.0) | 3.7 (0.2) | 4.7 (0.5) | **4.2** (0.0) | **5.0** (0.0) | +7% | 7 |
| ZORK3* | 0.3 (0.0) | 4.3 (0.5) | 0.5 (0.1) | **5.0** (0.0) | 3.0 (0.3) | **5.0** (0.0) | 3.8 (0.4) | 4.7 (0.5) | **4.2** (0.1) | **5.0** (0.0) | +6% | 7 |
| PENTARI | 43.4 (4.5) | 58.3 (2.4) | 29.8 (14.1) | 46.7 (6.2) | 37.4 (7.0) | 46.7 (11.8) | 43.4 (1.7) | **60.0** (0.0) | **45.5** (4.3) | **60.0** (0.0) | +3% | 70 |
| PENTARI* | 45.6 (1.9) | 58.3 (2.4) | 34.5 (7.5) | 53.3 (6.2) | 33.4 (6.9) | 46.7 (6.2) | 43.4 (0.4) | **60.0** (0.0) | **49.6** (1.3) | **60.0** (0.0) | +6% | 70 |
| DETECTIVE | 289.9 (0.1) | 320.0 (8.2) | 289.5 (0.4) | 323.3 (4.7) | 269.3 (14.8) | **346.7** (4.7) | 296.0 (9.0) | 336.7 (12.5) | **312.2** (10.3) | 340.0 (8.2) | +4% | 360 |
| BALANCES | **10.0** (0.0) | **10.0** (0.0) | 9.9 (0.0) | **10.0** (0.0) | **10.0** (0.0) | **10.0** (0.0) | 9.6 (0.1) | **10.0** (0.0) | **10.0** (0.0) | **10.0** (0.0) | 0% | 51 |
| BALANCES* | 14.1 (0.4) | 25.0 (0.0) | 12.5 (1.6) | 25.0 (0.0) | 10.0 (0.1) | 18.3 (2.4) | 21.9 (0.4) | 25.0 (0.0) | **24.0** (0.3) | **26.7** (2.4) | +4% | 51 |
| LIBRARY | 17.3 (0.7) | **21.0** (0.0) | 17.0 (0.2) | **21.0** (0.0) | 16.2 (1.4) | **21.0** (0.0) | 18.8 (0.4) | **21.0** (0.0) | **20.9** (0.1) | **21.0** (0.0) | +3% | 30 |
| LIBRARY* | 24.8 (0.6) | **30.0** (0.0) | 24.7 (0.4) | **30.0** (0.0) | 24.2 (1.4) | **30.0** (0.0) | 26.1 (0.4) | **30.0** (0.0) | **28.5** (0.3) | **30.0** (0.0) | +8% | 30 |
| DEEPHOME | 57.9 (0.4) | 68.7 (0.5) | 44.8 (19.9) | 76.0 (5.0) | 1.0 (0.0) | 1.0 (0.0) | 46.3 (9.0) | 60.7 (13.2) | **75.7** (5.0) | **93.7** (5.6) | +6% | 300 |
| DEEPHOME* | 58.8 (0.1) | 68.0 (0.8) | 58.9 (0.2) | 72.7 (3.8) | 1.0 (0.0) | 1.0 (0.0) | 52.6 (0.4) | 70.0 (0.8) | **77.7** (2.1) | **92.3** (3.3) | +6% | 300 |
| ENCHANTER | **46.1** (11.1) | 70.0 (21.2) | 46.0 (3.6) | **73.3** (8.5) | 25.8 (8.5) | 36.7 (14.3) | 43.4 (18.9) | 53.3 (23.6) | 34.7 (21.2) | 36.7 (23.6) | -3% | 400 |
| ENCHANTER* | 42.0 (1.2) | **66.7** (2.4) | 44.2 (18.3) | 63.3 (30.6) | 26.8 (1.9) | 38.3 (4.7) | 24.3 (10.8) | 28.3 (11.8) | **52.0** (23.1) | **66.7** (33.0) | +2% | 400 |
| DRAGON | -3.7 (0.4) | 8.0 (0.0) | -2.3 (0.5) | 8.7 (1.7) | 3.2 (1.6) | 8.0 (0.0) | 40.7 (0.0) | **126.0** (0.0) | **96.7** (1.1) | **127.0** (0.0) | 0% | 25 |
| OMNIQUEST | 8.2 (0.1) | 10.0 (0.0) | 9.9 (0.0) | **13.3** (2.4) | 9.3 (0.7) | 10.0 (0.0) | 8.6 (0.1) | 10.0 (0.0) | **11.6** (1.3) | **13.3** (2.4) | +3% | 50 |
| **Avg. Norm Score** | 29.5% (29.8) | 48.8% (28.8) | 28.4% (27.2) | 51.8% (27.3) | 29.7% (25.6) | 44.5% (32.1) | 49.2% (30.4) | 58.6% (33.6) | 56.3% (28.1) | 64.0% (28.6) | 5.8% (4.1) | 100% |

**Table 4: Full Deterministic Results.** Standard deviations are in parentheses. Scores are averaged across 3 seeds. Note that the average normalized scores only take into account the games listed in Table 1. Baselines were rerun with the latest Jericho version.

| Games | DRRN Avg | DRRN Max | INV-DY Avg | INV-DY Max | RC-DQN Avg | RC-DQN Max | XTX-Uniform Avg | XTX-Uniform Max | XTX (ours) Avg | XTX (ours) Max | Δ (%) | Game Max |
|---|---|---|---|---|---|---|---|---|---|---|---|---|
| ZORK1 | 41.3 (3.2) | 55.7 (3.3) | 36.9 (2.4) | 85.7 (14.8) | 40.3 (1.6) | 53.0 (0.0) | 31.2 (1.1) | 48.0 (5.0) | **67.7** (8.0) | **143.0** (10.7) | +8% | 350 |
| ZORK3 | 0.2 (0.0) | 4.3 (0.5) | 0.7 (0.2) | **5.0** (0.0) | **2.7** (0.0) | **5.0** (0.0) | 1.8 (0.1) | 4.0 (0.0) | 2.7 (0.4) | **5.0** (0.0) | 0% | 7 |
| ZORK3* | 0.2 (0.0) | 4.0 (0.0) | 0.4 (0.3) | 4.7 (0.5) | **2.7** (0.1) | 4.7 (0.5) | 2.3 (0.5) | 4.3 (0.5) | 2.6 (0.6) | **5.0** (0.0) | -1% | 7 |
| PENTARI | 42.3 (0.8) | **60.0** (0.0) | 28.9 (8.5) | 45.0 (0.0) | 31.2 (3.9) | 38.3 (11.8) | 38.4 (1.3) | **60.0** (0.0) | **48.2** (0.4) | **60.0** (0.0) | +8% | 70 |
| PENTARI* | 38.2 (3.6) | **60.0** (0.0) | 37.5 (8.0) | 55.0 (7.1) | 33.3 (6.0) | 41.7 (10.3) | 38.8 (0.4) | **60.0** (0.0) | **47.3** (0.4) | **60.0** (0.0) | +12% | 70 |
| DEEPHOME | 58.2 (0.6) | 72.0 (5.7) | 58.2 (0.5) | 72.7 (2.5) | 1.0 (0.0) | 1.0 (0.0) | 48.0 (10.1) | 62.0 (14.2) | **73.9** (4.3) | **99.3** (13.9) | +5% | 300 |
| DEEPHOME* | 43.0 (20.0) | 65.7 (3.3) | 58.4 (0.5) | 73.0 (1.4) | 1.0 (0.0) | 1.0 (0.0) | 50.7 (2.3) | 69.3 (0.9) | **70.9** (2.7) | **96.0** (7.8) | +4% | 300 |
| ENCHANTER | 41.0 (0.6) | **71.7** (9.4) | 38.9 (14.5) | 63.3 (30.6) | 25.0 (4.0) | 30.0 (7.1) | 32.1 (10.9) | 53.3 (23.6) | **46.2** (18.9) | 51.7 (22.5) | +1% | 400 |
| ENCHANTER* | 42.0 (18.5) | 56.7 (27.2) | 34.5 (10.3) | 53.3 (23.6) | 27.1 (2.7) | 43.3 (8.5) | 30.2 (9.1) | 45.0 (20.4) | **44.8** (19.4) | **58.3** (27.8) | +1% | 400 |
| **Avg. Norm Score** | 18.9% (18.2) | 39.0% (28.1) | 19.7% (17.5) | 41.5% (26.0) | 20.9% (18.6) | 30.5% (27.1) | 24.3% (17.9) | 39.1% (29.6) | 31.8% (19.8) | 48.9% (26.0) | 4.8% (4.7) | 100% |

**Table 5: Full Stochastic Results.** Standard deviations are in parentheses. Scores are averaged across 3 seeds. Note that the average normalized scores only take into account the games listed in Table 2. Baselines were rerun with the latest Jericho version.

## A.6 Aggregate Metrics & Performance Profiles

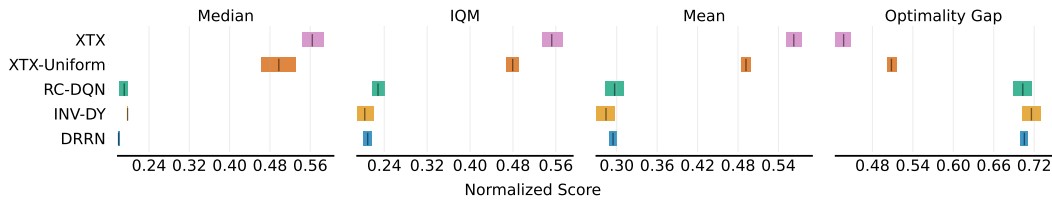

**Figure 16: Aggregate metrics** with 95% CIs for the deterministic games listed in Table 1, following Agarwal et al. (2021). The CIs use percentile bootstrap with stratified sampling.

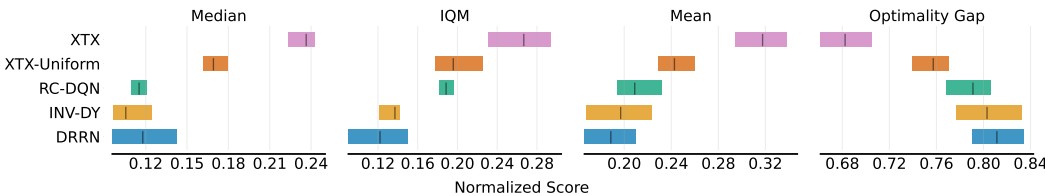

**Figure 17: Aggregate metrics** with 95% CIs for the stochastic games listed in Table 2, following Agarwal et al. (2021). The CIs use percentile bootstrap with stratified sampling.

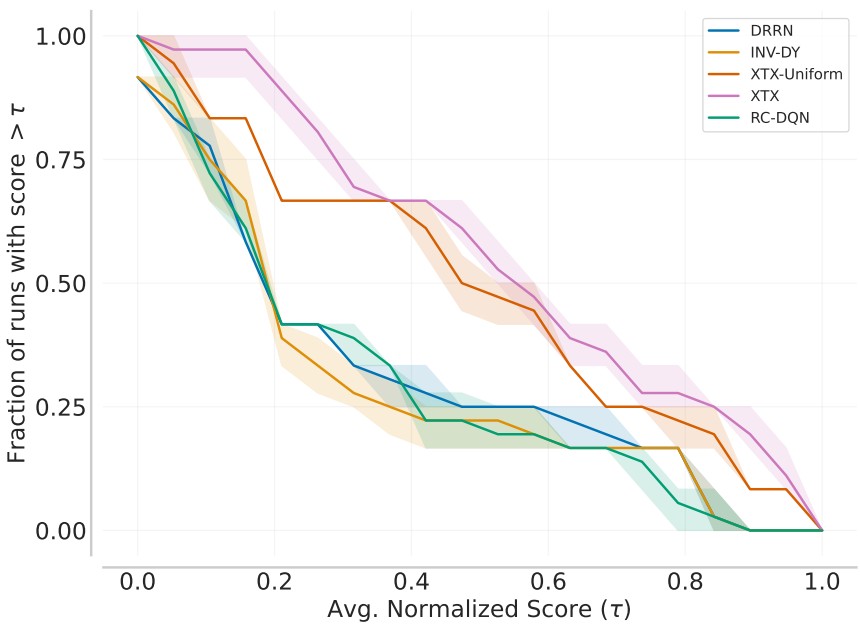

Figure 18: **Performance profiles** based on score distributions for the deterministic games listed in Table 1, following Agarwal et al. (2021). Shaded regions show pointwise 95% confidence bands based on percentile bootstrap with stratified sampling.

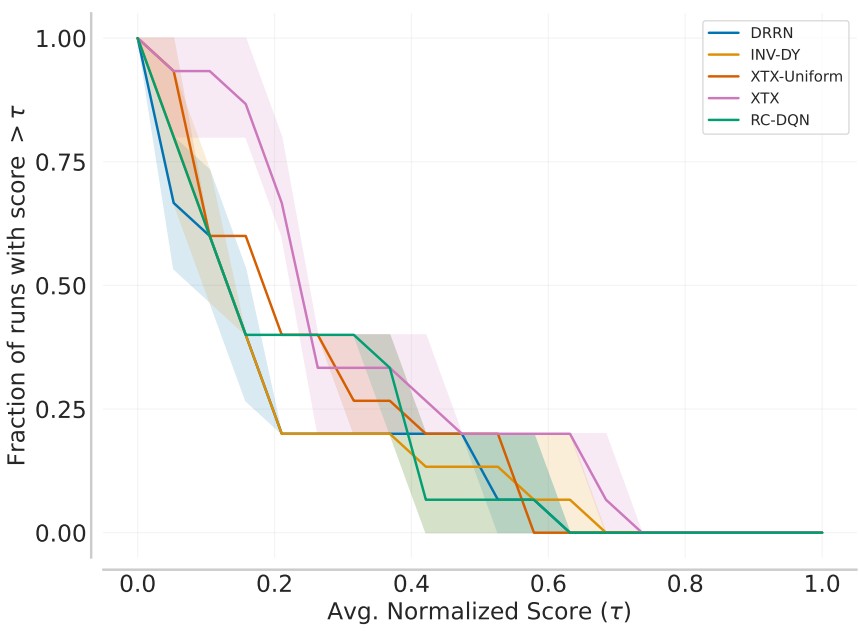

Figure 19: **Performance profiles** based on score distributions for the stochastic games listed in Table 2, following Agarwal et al. (2021). Shaded regions show pointwise 95% confidence bands based on percentile bootstrap with stratified sampling.

