# OpenReview forum: "Multi-Stage Episodic Control for Strategic Exploration in Text Games"
_ICLR.cc/2022/Conference — ICLR 2022 Spotlight_

### Official Review · Reviewer_Exgo · 2021-11-01

**Correctness:** 3
**Technical Novelty And Significance:** 2
**Empirical Novelty And Significance:** 3
**Recommendation:** 6
**Confidence:** 5

**Main Review:**

**Strengths**

1. The disentanglement of exploration and exploitation makes sense. The phase-alternating pipeline is nicely designed.
2. The paper is clearly written, it is relatively easy to understand how the model look like (although the intuition of each component isn't too clear).
3. The set of ablation experiments in Section 4.2 are well designed.

**Questions and concerns**

1. What's the reason of choosing this subset of 12 games? While the list seems to cover a wide range of game difficulties, but why not using the entire Jericho suite?
2. The authors cited the INV-DY agent (Yao et al., 2021) in their Section 3.1.3, and actually, if I understand correctly, the entire Section 3.1.3 is describing Yao et al.'s model, without any new contribution. Why do not the authors compare their agent with INV-DY in result tables?
3. In Section 3.1.PHASE 1, the authors describe two criteria that switch the agent to exploration phase. Can the authors elaborate on the second criterion, what does it mean if the number of steps in an episode equal to the longest of the $k$ sampled trajectories? If an agent moves back and forth between two locations, which may result a super long steps, but this behavior is not necessarily desired.
4. In Section 3.1.2, Sampling trajectories, the authors describe the way they use to sample trajectories. However, to my understanding, the loss shown in Eqn. 5 is a game-step-wise loss. Does the authors also sample game steps from the sampled trajectories (if so, how?), or they compute this loss on all game steps within the sampled trajectories?
5. In the paragraph under Eqn. 2, the authors mentioned that "Note that the action distribution over actions $a$ induced by $\pi_{inv-dy}$ is conditioned only on the current observation $o$". However, according to Eqn. 6, it is also conditioned on $o'$, which is the next observation, i.e., $o_{t+1}$.



**References**
1. Keep CALM and explore: Language models for action generation in text-based games. Shunyu Yao, Rohan Rao, Matthew Hausknecht, and Karthik Narasimhan. EMNLP 2020.
2. Reading and acting while blindfolded: The need for semantics in text game agents. Shunyu Yao, Karthik Narasimhan, and Matthew Hausknecht. NAACL 2021.


---------------------------- Nov 29, 2021
We had a good discussion among reviewers, let me give the authors some update.

**1.  Increased my score to 6.** This is because the authors have somewhat addressed my comments, I'm relatively satisfied. There are a few concerns remaining, as listed below:

a) on modelling novelty: the novel components are only a) the sampling strategy in exploitation phase and b) the two-phase pipeline. It was a bit weird to (almost) "copy and paste" a subsection from a prior work into the main body of this submission, which may confuse readers by giving a false message about the contribution. However, if other co-reviewers are fine with it, I'm fine too.

b) After a few paper updates, the main results (in Table 1) is only marginally higher than prior work. The authors can add more discussion addressing this in their cameraready.

**2.  We recommend the authors to remove the Dragon row from the result tables** (or rerun when the Jericho team fixes the bug):

As Reviewer PsKh find out, the proposed agent's scores exceed the max score on that game.

I happen to know some core Jericho contributors, and we tested the [Dragon game](https://ifdb.org/viewgame?id=sjiyffz8n5patu8l).

Usually, when reaching the goal, this will pop up:

```
Dragon's Treasure Store

The Dragon's secret hoard is open before you. By the flickering light of your little candle, you can make out a heaps of treasure stacked untidily around the floor.
You can see piles of gold and heaps of jewels, many rising higher than the top of your head. The Dragon has told you it has no use for the treasure and it is now yours.

You are rich beyond your wildest dreams!

*** You have won ***

In that game you scored 25 out of a possible 25, in 101 turns.

Would you like to RESTART, RESTORE a saved game, UNDO your last move, give the FULL score for that game or QUIT
```

In that game, the scoring function works like this:
```
1 for buying the box
1 for finding the screwdriver
2 for finding the candle
1 for finding the matches
1 for for opening the castle door
1 for building the hand-glider in the right place
2 for getting the sword/booklet
1 for escaping from the tower using the hang-glider
2 for killing the Troll to get the horn
5 for talking to the Troll to get the horn
2 for killing the dragon
5 for charming the dragon instead of killing him
5 for finding the treasure

= 25 points maximum total (e.g., multiple ways to get the horn)

(minus 2 points for each RESCUE or 'dead' recovery)
```
Given this -2 points for each RESCUE action, an agent can get negative total points. Because the original game did some *short* to *unsigned char* converting, this caused underflow (-128 vs. 128).

This may because the author of the Dragon game (in 2003) didn't expect machines to play his game, because most humans will give up playing before reaching this underflow point :)

So the weird numbers are not the authors' problem. As I mentioned, they can either remove that row, or to rerun whenever Jericho fixes that.


**Summary Of The Paper:**

In this paper, the authors propose eXploit-Then-eXplore (XTX), a training strategy for agent solving human-generated text-based games.

XTX consists of two training phases:

1. In the exploitation phase, the agent samples high quality experiences (in terms of score and trajectory length) from its replay buffer. Using the sampled trajectories, an action generation module is trained. At a certain game step $t$, the action generation module takes the observation $o_t$, as well as the two most recent past actions $a_{t-1}$ and $a_{t-2}$ as input, and generates the new action $a_t$ in a word-by-word auto-regressive manner. This process is referred as self-imitation by the authors.

2. In the exploration phase, in addition to the Q-learning loss as used in DRRN, the authors use two auxiliary losses to encourage the model to capture useful representations. First, the inverse dynamics loss $L_{inv}$ optimizes a module that predicts an action $a_t$ given two consecutive observations $o_t$ and $o_{t+1}$, where $o_{t+1}$ is resulted by $a_t$ given $o_t$. The second loss $L_{dec}$ is a regularizer that optimizes a module reconstructs an action $a_t$ from its encoding $f_a(a_t)$.

During training, the two phases take control in an (almost) alternate manner, however, there is a coefficient $\lambda$ controls the interpolation between the phases. The authors show that it is beneficial to not having the exploitation take control solely.

On a subset of games from the Jericho suite, the authors show their agent outperform prior works.


**Summary Of The Review:**

While I like this paper in general, my main concern is its novelty and contribution. As mentioned in my questions and concerns (Q2) above, the entire Section 3.1.3 is describing prior work (Yao et al., 2021), the "Learning from trajectories" part of Section 3.1.2 is describing another prior work (Yao et al., 2020). Actually, neither (Yao et al., 2021) nor (Yao et al., 2020) is compared in result table. As a consequence, to my understanding, the contribution of this paper is the two-phase pipeline and the sampling strategies in Section 3.1.2. I am not sure if this paper contains enough contributions to publish at ICLR.

Please correct me if I understood wrong.

---

> ### Author Response · Authors · 2021-11-16
> **Response to Reviewer Exgo**
>
> Thank you for your valuable feedback. Please feel free to ask any additional follow-up questions. For a clarification on our main contributions, novelty, and results, please refer to parts 1 - 4 in the General Response. In what follows we first address your comments about prior work, and then address your additional specific questions.
>
>
> #### **Comparison with prior work**
> Regarding your note about prior work, **we do compare with the inverse dynamics model from [1]** and apologize for not making this more clear in the paper. These results are included in the ablations (Figure 2, referred to as “pure inverse dynamics”). For now, **we will add 2 extra games for these ablations** for a total of 6 games, and we intend to include the full set of 12 games as soon as they finish running. Overall, we find the performance of the INV-DY agent [1] to be much worse than XTX.
>
> Note that **we do not feel a comparison with [2] is possible since they focus on a different setting**. Specifically, their work replaces the built-in Jericho valid action detection handicap with a GPT-2 based language model which generates action candidates. Our work does make us of this handicap. Hence, their model and our model make different assumptions. **One could of course swap the language model in CALM with the Jericho valid action handicap, but this just results in DRRN (which is better than CALM as shown in [2]), which we do compare with in the table**.
>
> For a further discussion of our work compared to prior text game methods and to other exploration methods, please refer to parts 1 & 3 of the General Response.
>
>
> Please find our answers to your numbered questions and concerns below (keeping the numbering consistent):
>
> #### **1. Why focus on a subset of Jericho?**
> We focused on a subset of games following prior work [1, 3, 4]. In addition, running the full set of games requires extensive computational resources and we wanted to ensure a sufficient number of seeds for each method. To give a sense for this, running all models on the current set of 12 games with 3 seeds already requires around 350 experiments (each taking anywhere from 1 - 3 days), not including the ablations. Nonetheless, we are currently running on additional games as well and hope to include these soon into the revised paper.
> #### **2. Comparison with INV-DY?**
> We agree that the inverse dynamics model from [1] is an important model to compare to. As noted above, these results were already included in the ablation section (Figure 2 on page 9, referred to as “pure inverse dynamics”). We will further clarify in the paper that this ablation is equivalent to the inverse dynamics model from [1].
> #### **3. Clarification of second switching criterion**
> The reason for waiting until the number of steps taken equals the longest of the $k$ sampled trajectories before switching phases is that **we want to allow the agent to be able to return to any of the $k$ sampled trajectories, including the longest one**. It is possible that the agent needs less steps to return to a particular part of the game space and hence, as you mentioned, wastes some steps because of this heuristic, but we empirically didn’t find this to be a problem. Additionally, to mitigate the effect of the agent moving back and forth between locations **we heavily bias sampling towards short trajectories**. Qualitatively, we found this significantly reduces noise (e.g. the agent moving back and forth). We will further clarify both of the points above in a revised version of the paper.
>
> #### **4. Clarification on loss in equation 5**
> Regarding the imitation learning on the sampled trajectories, we indeed perform the loss in equation 5 at every step in those trajectories, where a step is given by a pair $(c, a)$ where $c$ is the context (in our case the current observation and 2 past actions), and $a$ is the current action. We will further clarify this in a revised version of the paper.
>
> #### **5. Clarification on conditioning of $\pi_{\mathrm{inv-dy}}$**
> We apologize for the confusion around what $\pi_{\mathrm{inv-dy}}$ is conditioned on. $\pi_{\mathrm{inv-dy}}$ is the policy used during the second part of exploration and induces a distribution over the valid actions conditioned only on the current observation $o$. However, the encoders $f_o$ of $\pi_{\mathrm{inv-dy}}$ are updated throughout training using the auxiliary inverse dynamics loss presented in equation 6. Note that $o’$ is only used there for the MLP $g_{\mathrm{inv}}$ and the decoder $d$, both of which are only used for the inverse dynamics loss itself, not to score the actions during gameplay. Scoring the actions will be done with the MLP $q$ (not $g$, this is a typo) that implements the Q network, as defined in the Background paragraph of section 3. We will correct and clarify this in a revised version of the paper.
>
> (to be continued)

---

> > ### Author Response · Authors · 2021-11-16
> > **Response to Reviewer Exgo (continued)**
> >
> > #### **References**
> > [1] Yao, S., Narasimhan, K., & Hausknecht, M.J. (2021). Reading and Acting while Blindfolded: The Need for Semantics in Text Game Agents. NAACL.
> > [2] Yao, S., Rao, R., Hausknecht, M., & Narasimhan, K. (2020). Keep CALM and explore: Language models for action generation in text-based games. EMNLP.
> > [3] Jang, Y., Seo, S., Lee, J., & Kim, K. (2021). Monte-Carlo Planning and Learning with Language Action Value Estimates. ICLR.
> > [4] Ammanabrolu, P., Tien, E., Hausknecht, M.J., & Riedl, M.O. (2020). How to Avoid Being Eaten by a Grue: Structured Exploration Strategies for Textual Worlds. ArXiv, abs/2006.07409.

---

> > > ### Comment · Reviewer_Exgo · 2021-11-21
> > > **Thanks**
> > >
> > > Thanks for the response.
> > >
> > > In my opinion, INV-DY should not only be an ablation. It should be a baseline in Table 1. Because to my understanding, [1]* is one of the two phases in your framework, unless Section 3.1.3 in the submission has any significant difference from [1].
> > >
> > > Also, I'm quite confused why in the general response, neither [1] nor [2] is referred in the "Contribution" and "Comparison with other exploration methods" paragraph? To my understanding these two works are more than related. It seems to me that the authors are somewhat hesitant to exhibit the connection between this submission and [1][2]?
> > >
> > > To be clear, I recognize and appreciate the novel part of this work (the two-phase framework etc.), but I am confused by how [1][2] are presented in this work.
> > >
> > > Always, please correct me if I'm wrong.
> > >
> > > * re-using the above reference list in the authors' response.

---

> > > > ### Author Response · Authors · 2021-11-22
> > > > **Response #2 to Reviewer Exgo**
> > > >
> > > > Thank you again for your valuable feedback. Please find our responses to your concerns below.
> > > >
> > > > #### **Comparison with INV-DY**
> > > > While we do think it is valuable to have INV-DY as an ablation to show that XTX is a general method and can reduce to INV-DY when setting $\lambda = 1$, **we have also moved INV-DY in the main table of results (Table 1)** as you suggested. We will also add in Stochastic results (Table 2) once the runs are completed.
> > > >
> > > > #### **Connection of our work with INV-DY [1] and CALM [2]**
> > > > Sorry, we didn't mention [1] and [2] in our General Response since we intended for the General Response to mostly include common concerns and clarifications among various reviewers. We initially addressed your concerns about [1] and [2] in our previous individual response in this thread above, under "Comparison with prior work". **However, we agree it would be benefical to further clarify the relation of our submission to these works, as we fully acknowledge the connection of [1] and [2] to our work and consider them as valuable inspirations.** We briefly do this below and have also revised the paper accordingly (changes marked in blue in paper).
> > > >
> > > > The INV-DY model of [1] is equivalent to the second phase of our algorithm, but lacks an explicit distinction between exploration and exploitation. We clarified these points in the paper under the "Directed exploration in text-based games" paragraph in the related work (section 2), as well as in the "Novelty in comparison to prior algorithms" paragraph at the end of section 3.1.3.
> > > >
> > > > The work of [2] uses language models for exploration but focuses on a different setting where there is no handicap to provide valid actions, and instead the exploration is guided by a language model that *generates* the most likely action candidates. While the training of our exploitation policy $\pi_{\mathrm{il}}$ is inspired by this work, $\pi_{\mathrm{il}}$ in our work is used to mimick promising trajectories during gameplay instead of to generate action candidates. We added these points to the paper under the "Directed exploration in text-based games" paragraph in the related work (section 2), as well as at the end of section 3.1.2.
> > > >
> > > > We hope these changes address your comments and clear up any remaining confusion! If not, feel free to ask any additional follow-up questions or clarifications.
> > > >
> > > >
> > > > #### **References**
> > > > [1] Yao, S., Narasimhan, K., & Hausknecht, M.J. (2021). Reading and Acting while Blindfolded: The Need for Semantics in Text Game Agents. NAACL.
> > > > [2] Yao, S., Rao, R., Hausknecht, M., & Narasimhan, K. (2020). Keep CALM and explore: Language models for action generation in text-based games. EMNLP.

---

> > > > > ### Author Response · Authors · 2021-12-07
> > > > > **Thank you**
> > > > >
> > > > > Thank you for the valuable discussion. We will make sure to restructure the inverse dynamics part of our methods section in the final version of the paper, as well as add a footnote to the Dragon results explaining the broken score (thank you for investigating this!).

---

### Official Review · Reviewer_PsKh · 2021-11-03

**Correctness:** 4
**Technical Novelty And Significance:** 3
**Empirical Novelty And Significance:** Not applicable
**Recommendation:** 8
**Confidence:** 3

**Main Review:**

The main contribution is an exploration strategy with an in-episode switch from an exploitation policy to one aimed at exploration. This approach to combining exploration and exploitation is different from much of the existing literature, where typically a single policy is used throughout the episode, and often throughout training, that merges two reward signals. Since the switching policy in this paper is the element that looks most hardcoded, and therefore potentially brittle, it would be valuable to investigate a bit more whether a more flexible solution is also possible here. While different, Agent57 (whose predecessor NGU is cited) might offer inspiration here: it also uses multiple policies, and manages the switching with a learned (bandit) mechanism. A significant difference is that there the switching only happens between episodes, but a similar switching mechanism might be considered here within episodes nonetheless.

The in-episode switch is there to ensure that exploration happens at the edge of the known region of state space, where it is needed and meaningful. That is a very sensible thing for the agent to do, but there are other exploration strategies that effectively also do that, such as random network distillation (Burda et al., 2018) and inverse dynamics (Pathak et al., 2017), which the authors use to train their exploration policy. While the exploration region is less explicitly located at the edge of the known state space region in those algorithms than in this paper, the prediction errors that they rely on for intrinsic reward generation are more likely to occur at that edge. One question I have for the authors here is whether the inverse dynamics reward signal itself can be used to indicate when to switch from explore to exploit. In that case, the two-policy solution can be simplified again to a single policy that merges the two behaviours. I did not see this ablation in the paper, but I believe it would be a good thing to include.

Conversely, it would be valuable to see the performance of the strategy proposed here on other exploration benchmarks, such as the hard exploration games from the Atari suite (Bellemare et al., 2016). While I appreciate that text adventure games are in some ways different from their video counterparts, since they have a different observation space (language, not pixels) and action set (again language, not moves), they are still both RL environments, and general agents should be able to play both. Furthermore a game like Montezuma’s Revenge has a bottleneck aspect similar to the one that many text based games have, as well as the need for exploration on the frontier of the known region of state space. All in all it seems that the proposed strategy here could work on a wider range of environments than addressed in the paper. If that is not the case, it is still a valuable contribution, but if it is, it would be good to know.

A last comment: the agent proposed in the paper has another unusual feature in that its exploitation policy is trained only by self-imitation. While it is important to find the edge of the explored region of state space, and the self-imitation training regime can help with this, the XTX strategy can also be implemented with an exploitation policy that is trained in a more traditional way, with one of the many RL approaches available. Can the authors comment on why they chose the self-imitation approach instead?


**Summary Of The Paper:**

This paper introduces an agent with a built-in exploration strategy that is aimed at text adventure games, or more generally, environments with large action spaces and sparse rewards. The exploration strategy is constructed from two independent policies: one trained with self-imitation learning on successful trajectories, and one trained on an inverse dynamics intrinsic reward. The agent plays episodes by starting with the exploitation policy for a number of steps that depends on the experience collected up to that point, and then switching to the exploration policy. The paper is well-written, describes the contributions clearly, and places itself in the context of the existing literature on exploration. It includes results on a number of text exploration games from a recent benchmark, where it shows by and large a significant improvement relative to the baselines included.

**Summary Of The Review:**

The paper is well written, presents a marked improvement over the baselines provided (I’m not sufficiently familiar with the text adventure game literature to be certain those represent state of the art, but I will assume they do unless corrected), and provides an interesting approach to the exploration problem through the two-policy architecture. I recommend acceptance, but I also feel the paper could be strengthened by addressing the questions raised in the main review section.

---

> ### Author Response · Authors · 2021-11-16
> **Response to Reviewer PsKh**
>
> Thank you for your valuable feedback. Please feel free to ask any additional follow-up questions.
>
> #### **1. Improving the switching mechanism**
> Empirically we find the heuristic mechanism for switching between policies to work well, but we agree this could be a potential point of improvement, and appreciate the references towards related work on this topic such as Agent57. While we find the reviewer’s suggestion to switch based on the inverse dynamics signal a creative idea, this would require the setting of additional hyperparameters to determine the threshold of when to switch. In addition, the reason why our imitation learning policy is trained on multiple trajectories is so we can build a policy cover of the game space which allows returning even to earlier or different parts of the game where the inverse dynamics loss might be low. For example, this could happen if for a specific episode, the stochastic policy $\pi_{\mathrm{il}}$ decides to follow one of the lower scoring trajectories which brings the agent somewhere earlier in the game rather than at the frontier. Switching on inverse dynamics would not allow for this, and hence if the agent would have needed to fix something in the past to push the current frontier, it would be hard to do so.
>
> #### **2. XTX on other environments**
> As mentioned in part 3 of the General Response, we focus on the unique challenges of text games in this paper. However, we agree that it would be valuable to consider running XTX for other hard-exploration tasks like the ones you suggested, and leave this to future work.
>
> #### **3. On the use of self-imitation learning for exploitation**
> We chose the self-imitation approach versus more traditional RL approaches such as Q learning as **self-imitation is well-suited for fast, adaptive exploitation** – we train a new exploitation policy every epoch from $k$ newly sampled trajectories. We have previously tried using a Q-based approach but could not get this to stably work especially for longer trajectories.

---

### Official Review · Reviewer_TAdH · 2021-11-03

**Correctness:** 4
**Technical Novelty And Significance:** 3
**Empirical Novelty And Significance:** 3
**Recommendation:** 6
**Confidence:** 4

**Main Review:**

This paper is well motivated and most parts are well written, but the main method section is written to be difficult to follow. The results demonstrate empirical gains in the Jericho environment. However, the baselines consist only of simple algorithms without an exploration strategy. The detailed comments and questions are as follows:

1. In the experiment, the performance is compared with DRRN and MPRC-DQN, which lack exploration strategy. XTX seems to be an exploration method very similar to Go-Explore. Moreover, in the paper, Go-Explore and PC-PG are mentioned as the most closely related approaches, but they are excluded from the baseline algorithms. It would be better to demonstrate the results of them together.

2. (Page 5, section 3.1.2, sampling trajectories) It is hard to follow the explanation. Can it be understood as a kind of weighted behavior cloning? Moreover, I understand the motivation of biased sampling towards high scores, but don’t understand the motivation for the length. I think that a shorter trajectory length is not necessarily better. Can you give an intuitive explanation?

3. In the paper, policy $\pi_\text{il}$ is modeled as GPT-2, and policy $\pi_\text{inv-dy}$ is modeled as DRRN. Is there any reason why each policy is modeled differently? Especially, the policy $\pi_\text{il}$ is renormalized over the valid action set, is there any reason or advantage to learn the policy with GPT-2?

4. In the experiments, the results demonstrate XTX underperforms DRRN on ENCHANTER. Is there any intuitive explanation for this result? It would be better if a discussion about what characteristics in the ENCHANTER made the XTX not work would be added.



**Summary Of The Paper:**

This paper presents a new exploration algorithm, eXploit-Then-eXplore (XTX), for text-based games which require extensive exploration. The authors propose an algorithm that explicitly disentangles exploitation and exploration strategies within each episode. XTX first learns the exploitation policy that imitates the promising trajectories from past experiences, then uses exploration policy to discover the novel state-action spaces. Finally, the authors demonstrated the outperforming results in the Jericho environment.

**Summary Of The Review:**

This paper is well-motivated, written overall, and demonstrates state-of-the-art performance in the Jericho environment. However, there are relevant but missing baseline algorithms (Go-Explore, PC-PG) for the main table of experiments. I think the results of these algorithms should also be included in the main table, and I think this can further support the main arguments of the paper.

---

> ### Author Response · Authors · 2021-11-16
> **Response to Reviewer TAdH**
>
> Thank you for your valuable feedback. Please feel free to ask any additional follow-up questions.
>
> #### **1. Comparison with Go-Explore**
> We agree that Go-Explore is an important baseline to compare to, and previously had this result included as one of the ablations (Figure 2 on page 9, marked as "Go-Explore"). We apologize for this misplacement and will remove it from the ablation section and add it as a baseline in the main result tables (Tables 1 & 2 on pages 8 - 9), which will also include additional experiments we ran other than the original four games from the ablation section.
>
> Furthermore, **although we call one of our ablations as being similar to "Go-Explore", it really is XTX with uniform exploration**. This distinction is important since ***XTX-Uniform* avoids using additional memory archives as well as the complexities of goal-based policies for returning to specific states (e.g. specifying intermediate rewards etc.)**. Instead, it simply uses trajectories from the replay buffer and performs self-imitation learning on those to learn a policy cover over the promising parts of the game space. Nonetheless, XTX-Uniform is conceptually the closest variant to Go-Explore that allows for a fair comparison under the setup we consider. Note that this version of XTX-Uniform is also sufficiently close to PC-PG (i.e. it uses imitation learning to implicitly build a policy cover) so that we did not deem it necessary to have separate results for PC-PG.
>
> When comparing XTX with XTX-Uniform, we noticed that the improvement is most prominent on Zork1 in the deterministic case, **but additional experiments in the stochastic setup also show significant improvements of XTX over XTX-Uniform on Zork1, Zork3, and Deephome**. The comparison for the additional stochastic results can be found below (and will be included in Table 2 on page 9). As one can see, with max scores of 143, 5, and 91.7 on Zork1, Zork3, and Deephome respectively, XTX pushes significantly past the max scores of XTX-Uniform of 48, 4.3, and 69.3.
>
> | **Stochastic Setup** | **XTX-Uniform** |      | **XTX (Ours)** |       |          |
> |------------------|-------------|------|------------|-------|----------|
> | **Games**            | **Avg**         | **Max**  | **Avg**        | **Max**   | **Game Max** |
> | Zork1            | 31.2        | 48.0 | **67.7**       | **143.0** | 350      |
> | Zork3*           | 2.3         | 4.3  | **2.6**        | **5.0**   | 7        |
> | Pentari*         | 38.8        | **60.0** | **46.2**       | **60.0**  | 70       |
> | Deephome*        | 50.7        | 69.3 | **75.4**       | **91.7**  | 300      |
> | Enchanter*       | 30.2        | **45.0** | **33.8**       | 36.7  | 400      |
>
>
> In addition, we note that replacing the random exploration policy with inverse dynamics never significantly hurts performance, and can help quite a lot for a few games. This shows that inverse dynamics can be a useful exploration augmentation in bumping past bottlenecks (as previous work [1] has hinted) that other models have been unable to get past, and that the benefit is not specific to one game in one setup.
>
> #### **2. Sampling and learning from trajectories**
> Regarding section 3.1.2, it is indeed performing behavioral cloning on the context-action pairs (c, a) that are uniformly sampled from the trajectories in the trajectory buffer $\mathcal{B}$ (see "Sampling trajectories" in section 3.1.2). The reason we have a bias towards sampling shorter trajectories is simple. From analyzing some sample trajectories, it became clear to us that among trajectories that achieve the same score, **the shorter ones typically waste less time performing meaningless actions** (e.g. “pick up sword”, “drop sword”, “pick up sword”, etc.), and hence allow for more steps of exploration over the course of training. We will clarify this intuition in an updated version of the paper as well.
>
> (to be continued)

---

> > ### Author Response · Authors · 2021-11-16
> > **Response to Reviewer TAdH (continued)**
> >
> > #### **3. Modeling differences between $\pi_{\mathrm{inv-dy}}$ and $\pi_{\mathrm{il}}$**
> > The reason for having a separate imitation learning policy and Q-based policy augmented with inverse dynamics is that **these policies serve different purposes**. The imitation learning policy is fit on several promising trajectories (according to our sampling procedures outlined in section 3.1.2) and its purpose is to learn a global policy cover over promising parts of the game space that the agent can return to. On the other hand, the Q-based inverse dynamics policy is designed to perform efficient local exploration. Combining these models into one policy while maintaining XTX’s strength of explicitly disentangling exploitation and exploration is therefore nontrivial, but we encourage future work to explore this. Finally, the reason we choose a GPT-2 architecture for the imitation learning policy is because it is simple and has been shown to be effective for this kind of task in prior work [1].
> >
> > #### **4. Analysis of Enchanter results**
> > We agree it is interesting DRRN is able to significantly surpass XTX on Enchanter. After some analysis, we found the XTX agent is getting stuck at the point where it has to use a specific spell to open a gate, where the spell needs to be learned or carried with on a scroll found several steps ago (this is a bottleneck where the score bumps from 20 to 40). The XTX agent is rarely able to have the scroll at the gate and use the spell, while the DRRN agent can consistently do this. **Our current hypothesis for this is that DRRN is given a larger number of exploration steps per episode.** Specifically, we found that the first +20 reward in the game always happens at the very first step, so essentially all trajectories in XTX's trajectory buffer $\mathcal{B}$ are length 1, causing the number of environment steps per episode to always be 51, which is significantly lower than the 100 steps of exploration every episode that DRRN is able to use. To verify this, **we ran XTX with 100 steps of exploration instead of 50 (i.e. R = 100 in Algorithm 1), and already found the average performance to be slightly better than DRRN** (DRRN avg: 42.0, DRRN max: 66.7, XTX avg: 43.6, XTX max: 55, after ~80k steps of training). This shows that there is no fundamental flaw or reason why XTX underperformed on Enchanter; simply tuning a hyperparameter closed the gap.
> >
> > We will add this analysis to the results section of the paper as well.
> >
> > #### **References**
> > [1] Yao, S., Rao, R., Hausknecht, M., & Narasimhan, K. (2020). Keep CALM and explore: Language models for action generation in text-based games. EMNLP.

---

> > > ### Comment · Reviewer_TAdH · 2021-11-29
> > > **Response to rebuttal**
> > >
> > > Thank you for providing the rebuttal to respond to my questions and concerns. Most of my comments and questions have been addressed. My biggest concern was that there are relevant but missing baseline algorithms (Go-Explore, PC-PG), but it has been addressed in the author's response and revision. However, after the revision including the updated results, I am concerned that the results of the updated baseline XTX-Uniform (i.e. Go-Explore) seem to be similar to the proposed method on most games without Zork1.
> > >
> > > In summary, I think that the updated results seem quite marginal but I think that the paper is well-motivated and well written. So, I will keep my score.

---

> > > > ### Author Response · Authors · 2021-12-07
> > > > **Thank you**
> > > >
> > > > Thank you for the valuable feedback and discussion!

---

### Official Review · Reviewer_186e · 2021-11-05

**Correctness:** 4
**Technical Novelty And Significance:** 3
**Empirical Novelty And Significance:** 4
**Recommendation:** 8
**Confidence:** 3

**Main Review:**

Pros:

The paper is generally well-written and easy to follow.
The novelty of XTX is clearly elaborated.
The method surpasses the existing method with a large margin on text-based games. The ablation studies show the individual components introduced by XTX can bring improvements.

Cons:

One weakness of the paper is these experiments did not clarify why the novel part of XTX (i.e. exploration with novelty bonus on the frontier) is helpful over random actions. The paper hypothesizes that novelty bonuses can encourage the agent to select promising actions in large action spaces. However, the ablation study (Figure 2) casts doubts on this hypothesis. XTX brings significant improvements over Go-explore in Zork1 but not other games. The difference doesn't seem to be correlated to the size of action spaces.

Questions:

I don't fully get why the method is motivated to solve the problems with large action space. How can an agent receive a novelty bonus if it did not enter that novel state by trying random actions? Do the authors assume the generalization of the neural network plays a key role here?

Other Suggestions:

The author might want to try other hard-exploration tasks. For example, minigrid or maze can be tested, if not Atari games like Montezuma Revenge. Since these are environments where existing exploration methods are developed, we can have a better understanding of how exactly XTX compares to other exploration algorithms, rather than the existing text-base game agent without directed exploration.



**Summary Of The Paper:**

Summary:

The paper proposes a multi-stage directed exploration algorithm, XTX. It first imitates previous high score trajectories and then switches to an exploration policy with novelty bonuses.
Conceptually, XTX is a method that extends Go-Explore which only acts randomly after reaching the frontier of familiar states.
The paper argues that with novelty bonuses, the agent will be encouraged to explore more promising actions. This can especially be helpful when the action space is large like text-based games.
Empirically, XTX shows strong performance on a large set of text-based games.

**Summary Of The Review:**

Reason for the Score:

The write-up and experiments in this paper are of good quality. The method itself is novel and the empirical finding in this paper might be particularly interesting for the audience of text-based RL. I have minor concerns author's claim about why this method works better than existing exploration algorithms while I'm happy to increase the score if they are addressed.

---

> ### Author Response · Authors · 2021-11-16
> **Response to Reviewer 186e**
>
> Thank you for your valuable feedback. Please feel free to ask any additional follow-up questions.
>
> #### **1. Comparison with Go-Explore**
> We apologize for causing confusion when comparing with Go-Explore. **Although we call one of our ablations as being similar to "Go-Explore", it really is XTX with uniform exploration**. This distinction is important since ***XTX-Uniform* avoids using additional memory archives as well as the complexities of goal-based policies for returning to specific states (e.g. specifying intermediate rewards etc.)**. Instead, it simply uses trajectories from the replay buffer and performs self-imitation learning on those to learn a policy cover over the promising parts of the game space. Nonetheless, XTX-Uniform is conceptually the closest variant to Go-Explore that allows for a fair comparison under the setup we consider. To make this clearer, we will remove it from the ablation section and add it as a baseline in the main result tables (Tables 1 & 2 on pages 8 - 9), which will also include additional experiments other than the original four games from the ablation section.
>
> When comparing XTX with XTX-Uniform, we noticed that the improvement is most prominent on Zork1 in the deterministic case, **but additional experiments in the stochastic setup also show significant improvements of XTX over XTX-Uniform on Zork1, Zork3, and Deephome**. The comparison for the additional stochastic results can be found below (and will be included in Table 2 on page 9). As one can see, with max scores of 143, 5, and 91.7 on Zork1, Zork3, and Deephome respectively, XTX pushes significantly past the max scores of XTX-Uniform of 48, 4.3, and 69.3.
>
> | **Stochastic Setup** | **XTX-Uniform** |      | **XTX (Ours)** |       |          |
> |------------------|-------------|------|------------|-------|----------|
> | **Games**            | **Avg**         | **Max**  | **Avg**        | **Max**   | **Game Max** |
> | Zork1            | 31.2        | 48.0 | **67.7**       | **143.0** | 350      |
> | Zork3*           | 2.3         | 4.3  | **2.6**        | **5.0**   | 7        |
> | Pentari*         | 38.8        | **60.0** | **46.2**       | **60.0**  | 70       |
> | Deephome*        | 50.7        | 69.3 | **75.4**       | **91.7**  | 300      |
> | Enchanter*       | 30.2        | **45.0** | **33.8**       | 36.7  | 400      |
>
> In addition, we note that replacing the random exploration policy with inverse dynamics never significantly hurts performance, and can help quite a lot for a few games. This shows that inverse dynamics can be a useful exploration augmentation in bumping past bottlenecks (as previous work [1] has hinted) that other models have been unable to get past, and that the benefit is not specific to one game in one setup.
>
> #### **2. Intuition behind inverse dynamics**
> We apologize for the confusion around the relation of inverse dynamics to large action spaces. Let us clarify two intuitions behind our use of inverse dynamics to tackle the large, dynamic action spaces. **The first intuition is similar to visitation-based bonuses, where once some actions are visited more, their bonus (i.e. prediction loss) will decrease, boosting the likelihood of other less-visited actions.** Consider an agent entering a new room. Assuming a uniform exploration policy (e.g. Go-Explore), the agent will keep trying *all* actions uniformly over time every time it enters this room, even ones that have already been found to be useless (e.g. “wait”). However, for inverse dynamics, the bonus for visited actions will go down over time since the prediction error will be lower. Inverse dynamics thus spends less time on useless actions and more time on novel, less-visited actions, making the agent more efficiently bias its search in large action spaces. Note that one could also consider exploring with a pure Q-based policy, but this would also give problems because of the *dynamic nature* of the action space. For completely new actions like "Odysseus" (see Figure 1), the Q function will likely not give a good estimate, while inverse dynamics will at least incentivize to explore this action since it will likely give a high prediction error.
>
> (to be continued)

---

> > ### Author Response · Authors · 2021-11-16
> > **Response to Reviewer 186e (continued)**
> >
> > **The second intuition is that even for the first step of choosing a new action to take (i.e. before any bonus has been received), the neural network might generalize through learning of past inverse dynamics bonuses to what new actions would look like** and hence *identify* novel actions before having tried them once. For example, through experience, the Q-network (which estimates future discounted return + bonus) might have learned "drop [item]" has a very low bonus as the transition is easy to predict (in the game, dropping items is quite common), thus guiding the agent to explore novel actions more. **Note that language semantics in text games is key to such generalization,** since the Q network can predict similar bonuses for semantically similar actions (e.g. "drop [item]" example above), something not possible in MDPs with more abstract action spaces.
> >
> > #### **3. Main contribution of the paper**
> > Please refer to parts 1 & 2 of the General Response for a detailed discussion of this.
> >
> > #### **4. XTX on other environments**
> > As mentioned in part 3 of the General Response, we focus on the unique challenges of text games in this paper. However, we agree that it would be valuable to consider running XTX for other hard-exploration tasks like the ones you suggested, and leave this to future work.
> >
> > #### **References**
> > [1] Yao, S., Narasimhan, K., & Hausknecht, M.J. (2021). Reading and Acting while Blindfolded: The Need for Semantics in Text Game Agents. NAACL.

---

> > > ### Comment · Reviewer_186e · 2021-11-21
> > > **Score Increased**
> > >
> > > Thanks for the additional experiments and detailed explanation of the motivation.
> > > Most of my concerns have been addressed so I increased the score.

---

> > > > ### Author Response · Authors · 2021-12-07
> > > > **Thank you**
> > > >
> > > > Thank you for the valuable feedback and discussion!

---

### Author Response · Authors · 2021-11-16
**General Response**

We thank all reviewers for their constructive feedback and comments. Below we restate our motivation and contribution, how we differ from prior approaches, and the significance of our results. We have also responded individually to each reviewer's comments.

1. **Motivation**: Text adventure games have been a testbed for building agents that can act based on textual knowledge [1]. Previous methods that tackle these games usually focus on language challenges and employ a single policy to balance exploration and exploitation [2, 3, 4, 5, 6]. However, we envision their potential also as a testbed for reinforcement learning. A key, unique challenge to solve text games is handling the combination of sparse rewards with large, dynamic action spaces (see Figure 1 for examples). Such a challenge has been under-explored by prior text game methods that focus more on the language side, and prior RL methods that are developed in domains with different assumptions. Hence we believe progress in this direction could be interesting and important for text game research and RL research in general.
2. **Contribution**: To tackle this challenge, we hypothesize it is essential to **more explicitly disentangle exploitation and exploration into two phases, with the first phase adapted for quick exploitation of sparse rewards, and the second phase designed for local exploration** that strategically handles the large, dynamic action space by exploring novel and under-explored actions. XTX incorporates both of these ideas, with the imitation learning policy $\pi_{\mathrm{il}}$ quickly bringing the agent to various promising parts of the game space, after which the Q-based inverse dynamics policy $\pi_{\mathrm{inv-dy}}$ performs the local exploration.
3. **Comparison with other exploration methods**: Outside of the text games domain, our approach shares some aspects with Go-Explore and PC-PG. However, the former relies on additional memory archives and goal-based policies to handle stochasticity, while the latter more explicitly builds a set of policies for visiting different regions of the state space. Furthermore, both of these methods behave randomly after reaching the frontier, while our method uses a Q-based policy combined with a novelty bonus which allows for the selection of more promising actions compared to a random policy in large, dynamic action spaces. **In this paper we specifically focus on the unique challenges of text games mentioned in (1)** and hence only evaluate on Jericho, but it would be interesting for future work to compare XTX with these methods on other environments such as Atari. **We did run additional experiments to include full results for XTX-Uniform, a Go-Explore-like baseline, in both the deterministic and stochastic setting**, which will be included in a revised version of the paper in Table 1 and Table 2, respectively.
4. **Significance of results**: By targeting the unique challenge in (1) with insights detailed in (2), XTX achieves substantial improvements over prior work. On the famous game of Zork1, XTX obtains a score of 103 in the deterministic setting, and 67 in the stochastic setting – large improvements over the previous SOTA of 40 and 41, respectively. More broadly, XTX **outperforms all baselines on 8 out of 12 games in the deterministic setting (Table 1), and 3 out of 5 games in the stochastic setting (Table 2)**, suggesting our method works consistently across different games and is robust against stochasticity.

#### **References**
[1] Hausknecht, M., Ammanabrolu, P., Côté, M. A., & Yuan, X. (2020, April). Interactive fiction games: A colossal adventure. In Proceedings of the AAAI Conference on Artificial Intelligence (Vol. 34, No. 05, pp. 7903-7910).
[2] He, M. (2016). Deep Reinforcement Learning with a Natural Language Action Space. In Proceedings of the 54th Annual Meeting of the Association for Computational Linguistics (Volume 1: Long Papers) (pp. 1621–1630). Association for Computational Linguistics.
[3] Narasimhan, R. (2015). Language Understanding for Text-based Games using Deep Reinforcement Learning. In Proceedings of the 2015 Conference on Empirical Methods in Natural Language Processing (pp. 1–11). Association for Computational Linguistics.
[4] Prithviraj Ammanabrolu and Matthew J. Hausknecht (2020). Graph Constrained Reinforcement Learning for Natural Language Action Spaces. In 8th International Conference on Learning Representations, ICLR 2020, Addis Ababa, Ethiopia, April 26-30, 2020. OpenReview.net.
[5] Adhikari, A., Yuan, X., Côté, M.A., Zelinka, M., Rondeau, M.A., Laroche, R., Poupart, P., Tang, J., Trischler, A., & , W. (2020). Learning Dynamic Knowledge Graphs To Generalize On Text-Based Games. In NeurIPS 2020.
[6] Xu, Y., Fang, M., Chen, L., Du, Y., Zhou, J., & Zhang, C. (2020). Deep reinforcement learning with stacked hierarchical attention for text-based games. Advances in Neural Information Processing Systems, 33.

---

### Author Response · Authors · 2021-11-18
**Paper Revision #1**

Dear reviewers,

We just uploaded a new revision of the paper including all deterministic and stochastic numbers on XTX-Uniform (a Go-Explore-like baseline), ablations for 2 more games, and further clarifications that were mentioned in the individual responses below (all changes are marked in red). Please don't hesitate to follow-up on any of our responses below or ask any questions you have.

Thank you!

---

### Author Response · Authors · 2021-11-21
**Thank you and gentle reminder for discussion.**

Dear reviewers,

Thank you again for your valuable and constructive feedback. They were insightful and have allowed us to improve the paper (see updated paper draft) through additional experiments and clarifications. Since we haven't heard anything since after we posted our responses, we encourage you to check out our General Response to all reviewers, as well as the individual responses where we address specific questions.

Please also let us know if there is any additional clarifications or experiments we can provide to show the merit of our paper.

Thank you and looking forward to hearing from all of you!

---

### Decision · Program_Chairs · 2022-01-20

**Decision:**

Accept (Spotlight)

**Comment:**

I thank the authors for their submission and active participation in the discussions. All reviewers are unanimously leaning towards acceptance of this paper. Reviewers in particular liked that the paper is well-written and easy to follow [186e,TAdH,Exgo], well motivated [TAdH], interesting [PsKh], novel [186e] and provides gains over baselines [186e,TAdH,PsKh] with interesting ablations [186e,Exgo]. I thus recommend accepting the paper and  I encourage the authors to further improve their paper based on the reviewer feedback.